# Maximum Coverage in Turnstile Streams
# with Applications to Fingerprinting Measures

**Alina Ene** [1]   **Alessandro Epasto** [2]   **Vahab Mirrokni** [2]   **Hoai-An Nguyen** [3]   **Huy L. Nguyen** [4]   **David P. Woodruff** [2,3]
**Peilin Zhong** [2]

## Abstract

In the maximum coverage problem we are given $d$ subsets from a universe $[n]$, and the goal is to output $k$ subsets such that their union covers the largest possible number of distinct items. We present the first algorithm for maximum coverage in the turnstile streaming model, where updates which insert or delete an item from a subset come one-by-one. Notably our algorithm only uses $\mathrm{poly} \log n$ update time. We also present turnstile streaming algorithms for targeted and general fingerprinting for risk management where the goal is to determine which features pose the greatest re-identification risk in a dataset. As part of our work, we give a result of independent interest: an algorithm to estimate the complement of the $p^{\text{th}}$ frequency moment of a vector for $p \geq 2$. Empirical evaluation confirms the practicality of our fingerprinting algorithms demonstrating a speedup of up to 210x over prior work.

## 1. Introduction

Maximum coverage is a classic NP-hard problem with applications including information retrieval (Anagnostopoulos et al., 2015), influence maximization (Kempe et al., 2015), and sensor placement (Krause & Guestrin, 2007). Given $d$ subsets of a universe containing $n$ items and a cardinality constraint $k \geq 0$, the goal is to output the $k$ subsets whose union covers the largest possible number of distinct items. A simple greedy algorithm using $O(knd)$ time and $O(nd)$ space achieves a tight $1 - 1/e$ relative approximation (assuming $\mathrm{P} \neq \mathrm{NP}$) by iterating for $k$ rounds and selecting the subset that maximizes the marginal gain at each step (Feige,

1998). However, its polynomial time and space complexity make it impractical for large-scale datasets. Our goal therefore is to develop more efficient algorithms for maximum coverage.

To this end, we present the first one-pass turnstile streaming algorithm for maximum coverage. In the turnstile streaming model, updates come one-by-one in a stream. Each update inserts or deletes an item from a subset. The goal is to process each update efficiently while only maintaining sublinear space and then output the answer at the end of the stream. Our algorithm uses sublinear space and notably only $\mathrm{poly} \log n$ update time to output a near-tight $(1 - 1/e - \varepsilon)$ approximation to the answer. Our algorithm is one-pass, meaning it takes only one pass over the stream of updates.

Our algorithm offers efficient time complexity even with direct access to the full input. The input can be provided as a sequence of updates, and the *total* runtime is near-linear in the number of items across the input subsets. In addition, allowing arbitrary deletions of items from subsets is critical for numerous applications, including the following extension to fingerprinting for dataset risk management.

We also develop turnstile streaming algorithms for targeted and general fingerprinting. The input consists of $n$ users and $d$ features, where each user has a value assigned for each feature. In targeted fingerprinting, the goal is to select $k$ features that minimize the number of users who share identical values with a given target user $u \in [n]$ at these features. In general fingerprinting, the goal is to select $k$ features that minimize the number of user pairs with matching values across those features. Here an update in the stream changes the value of a user at a feature.

Our fingerprinting algorithms fit into the broader privacy attack literature (Seonghun et al., 2023; Chia et al., 2019; Zhou et al., 2023) and is an extension of Chia et al. (2019) in privacy auditing and risk measurement. Fingerprinting, which identifies users based on unique attribute combinations in a dataset, poses a significant privacy risk. Our algorithms mitigate this risk by identifying the $k$ features most likely to allow adversaries to successfully fingerprint users. Previous work, except for Gulyás et al. (2016) (whose

---
[1]Boston University [2]Google Research [3]Carnegie Mellon University [4]Northeastern University. Correspondence to: Hoai-An Nguyen <hnnguyen@andrew.cmu.edu>, David P. Woodruff <dwoodruf@cs.cmu.edu>.

*Proceedings of the 42$^{nd}$ International Conference on Machine Learning*, Vancouver, Canada. PMLR 267, 2025. Copyright 2025 by the author(s).

algorithms we improve), has only measured the risk of entire datasets or fixed sets of features. In contrast, our time and space efficient algorithms are suited for real-time monitoring of the re-identification risks even as a dataset evolves. Additionally, targeted fingerprinting is a form of frequency estimation with potential applications in discovering heavy hitters (Bhattacharyya et al., 2016; Zhu et al., 2020).

Our algorithms are all linear sketches, powerful structures that compress the input matrix while time efficiently handling insertions and deletions. Unlike algorithms designed for specific models, linear sketches are applicable across a wide range of settings. Besides directly implying turnstile streaming algorithms, they are well-suited for distributed contexts. One popular application for linear sketches is in the coordinator model where there are $k$ distributed machines and one coordinator. The input is split into pieces, and each machine gets a piece. They can only communicate with the coordinator (and want to minimize communication), and the goal is to get a joint solution. Another application is in parallel computation where there are $k$ distributed machines who can communicate to compute a joint solution. The goal is to again minimize the amount of communication.

### 1.1. Related Work

There is an extensive body of work on the maximum coverage problem, and we only attempt to give an overview of the most relevant works. In the following, an $x$-relative approximation means that the number of distinct items covered by the $k$ subsets outputted by the algorithm is at least $x \cdot \textbf{OPT}$, where $\textbf{OPT}$ is the number of distinct items covered by the optimal solution. $\tilde{O}(\cdot)$ notation is used to suppress polylogarithmic factors in its argument.

McGregor & Vu (2019) provide a one-pass streaming algorithm[1] that outputs a $(1 - 1/e - \varepsilon)$- relative approximation for $\varepsilon \in (0, 1)$ in $\tilde{O}(d/\varepsilon^2)$ space. They consider the insertion-only set-arrival streaming model where each update reveals a subset and *all* the items it covers, and deletions are not supported. Bateni et al. (2017) give a one-pass $(1-1/e-\varepsilon)$ relative approximation algorithm that uses $\tilde{O}(d/\varepsilon^3)$ memory. They consider the insertion-only streaming model where deletions are not supported. They specifically provide an algorithm that carefully samples a number of item-subset relationships and then show that any $(\alpha)$-relative approximation on this smaller subsampled universe achieves a $(\alpha - \varepsilon)$-relative approximation for the original input. We use their sketch as a starting point.

There has also been work that achieves different approximation factors (Saha & Getoor, 2009; McGregor et al., 2021), in random-arrival streams (Warneke et al., 2023; Chakrabarti et al., 2024), with multiple passes (Chakrabarti

---

[1]See Appendix A.3 about the streaming model.

et al., 2024), and in more general submodular maximization in the insertion-only set-arrival streaming model (Badanidiyuru et al., 2014; Kazemi et al., 2019). In contrast to all of the above, our algorithm allows arbitrary insertions or deletions of items from subsets. Note that there are more lines of work in the model where each update is the insertion or deletion of an entire subset (rather than items from subsets which we consider) than listed above.

There has also been work on maximum coverage and submodular maximization in the dynamic model (Monemizadeh, 2020; Chen & Peng, 2022; Lattanzi et al., 2020). We note that these dynamic model algorithms do not achieve sublinear space and in some cases require exponential space. Moreover, they either have a worse update time than our algorithm or do not attain the same approximation quality.

### 1.2. Our Contributions

**Maximum Coverage Results.** We formalize the input as an $n \times d$ matrix $\textbf{A}$ where entry $A_{ij}$ is nonzero if item $i$ is in subset $j$ and 0 otherwise. An update takes the form $(i, j, \pm c)$, modifying $A_{ij}$ by adding or subtracting $c$.

**Theorem 1.1.** *Given $n \times d$ matrix $\textbf{A}$, integer $k \geq 0$, and $\varepsilon \in (0, 1)$, there exists a one-pass turnstile streaming algorithm using $\tilde{O}(d/\varepsilon^3)$ space and $\tilde{O}(1)$ update time that outputs a $(1 - 1/e - \varepsilon)$ relative approximation to maximum coverage with probability at least $1 - 1/d$.*

Note that the only dependence on $k$ in our space complexity is a $\text{poly} \log k$ factor which is suppressed by the $\tilde{O}$. Our space complexity matches that of Bateni et al. (2017) and, for constant $\varepsilon$, that of McGregor & Vu (2019). Additionally, several lower bounds exist. Assadi (2017) show that achieving a $(1 - \varepsilon)$-relative approximation in a constant number of passes requires $\Omega(d/\varepsilon^2)$ space. Assadi & Khanna (2018) show that even achieving a $n^{1/3}$ or $\sqrt{k}$ relative approximation in one pass with a sketch requires the sketch to have size $\Omega(d/k^2)$. McGregor & Vu (2019) shows that achieving better than a $(1 - 1/e)$- approximation in a constant number of passes requires $\Omega(d/k^2)$ space. Bateni et al. (2017) also show that any $(1/2 + \varepsilon)$-relative approximation multi-pass streaming algorithm requires $\Omega(d)$ space.

**Fingerprinting Results.** In fingerprinting, the input is a $n \times d$ matrix $\textbf{A}$ where $A_{ij}$ is the value of user $i$ at feature $j$. An update takes the form $(i, j, \pm c)$, modifying $A_{ij}$ by adding or subtracting $c$. We first reduce targeted fingerprinting to maximum coverage to improve upon the $O(nd)$ space and $O(knd)$ time algorithm of Gulyás et al. (2016). We note that Gulyás et al. (2016) achieve a $1 - 1/e$ approximation.

**Corollary 1.2.** *Given a $n \times d$ matrix $\textbf{A}$, target user $u \in [n]$, and $\varepsilon \in (0, 1)$, there exists a one-pass turnstile streaming algorithm using $\tilde{O}(d/\varepsilon^3)$ space and $\tilde{O}(1)$ update time that outputs a $(1 - 1/e - \varepsilon)$ relative approximation to targeted*

*fingerprinting with probability at least $1 - 1/d$.*

We also improve upon the $O(nd)$ space and $O(knd)$ time algorithm of Gulyás et al. (2016) (which achieves a $1 - 1/e$ approximation) for general fingerprinting. While general fingerprinting can be easily reduced to maximum coverage, it requires tracking all $\binom{n}{2}$ user pairs to determine whether they differ in value on a certain feature. Upon an update, potentially $O(n^2)$ user pairs could be affected, making it impractical to use a linear sketch under sublinear space constraints. As a result, we instead design our general fingerprinting algorithm by first designing the following framework for submodular maximization under cardinality constraints over certain functions. The proof of the following is deferred to Appendix C.

**Theorem 1.3.** *Take $N$ to be a set of $n$ items. Let $f : 2^N \to \mathbb{R}_{\geq 0}$ be a monotone, non-negative submodular function. Given input subsets $S_1, \ldots, S_d \in N$, we aim to maximize $f$ by selecting $k$ subsets. Also take $f$ to be linearly sketchable[2] with a $(1 \pm \gamma)$-relative approximation in $O(s)$ space. For $\varepsilon \in (0, 1)$, if we set $\gamma = \varepsilon/k$, then there exists a one-pass turnstile streaming algorithm which outputs a $(1 - 1/e - \varepsilon)$-relative approximation using $O(sk)$ space. The algorithm succeeds with probability at least $1 - 1/\operatorname{poly}(d)$ if querying the linear sketch of $f$ results in error at most $O(\frac{1}{\operatorname{poly}(d)k})$.*

We then instantiate this framework to solve general fingerprinting. To do this, we design a novel linear sketch[3] for estimating $n^p - F_p$ for $p \geq 2$ on a $n$ dimensional vector $\boldsymbol{x}$ where $F_p$ is the $p^{\text{th}}$ frequency moment. If we take $\mathcal{Z}$ to be the set of distinct values in $\boldsymbol{x}$ and $f_i$ to be the frequency of the $i^{\text{th}}$ distinct value in $\boldsymbol{x}$, then $F_p = \sum_{i \in \mathcal{Z}} f_i^p$. For example, for $\boldsymbol{x} = (1, 5, 5, 3, -2, 3, 3, 3)$ we have $F_p = 1^p + 2^p + 4^p + 1^p$. Here, updates are of the form $(i, \pm c)$ which performs $x_i \leftarrow x_i \pm c$.

**Theorem 1.4.** *Given a $n$-dimensional vector $\boldsymbol{x}$, constant $p \geq 2$, and $\gamma, \delta \in (0, 1)$, there exists a linear sketch of size $\tilde{O}(\gamma^{-\frac{2}{p-1}})$ that supports updates in $\tilde{O}(\gamma^{-\frac{2}{p-1}})$ time and outputs a $(1 \pm \gamma^{\frac{1}{p-1}})$ relative approximation to $n^p - F_p$ with probability at least $1 - \delta$.*

$n^p - F_p$ is the complement of the frequency moment of a dataset (with the error dependent on the complement), and we believe it to be of independent interest. Frequency moments have numerous applications. For example, $F_p$ for $p \geq 2$ can indicate the degree of the skew of data which is used in the selection of algorithms for data partitioning (DeWitt et al., 1992), error estimation (Ioannidis & Poosala, 1995), and more. See Alon et al. (1999) for a more in-depth discussion. There are also direct applications for the quantity $n^p - F_p$ such as our use of the sketch to instantiate

Theorem 1.3 solve general fingerprinting. The proof of the following is deferred to Appendix E.

**Theorem 1.5.** *Given a $n \times d$ matrix $\boldsymbol{A}$ and $\varepsilon \in (0, 1)$, there exists a one-pass turnstile streaming algorithm using $\tilde{O}(dk^3/\varepsilon^2)$ space and $\tilde{O}(k^3/\varepsilon^2)$ update time that outputs a $(1 - 1/e - \varepsilon)$ relative approximation to general fingerprinting with probability at least $1 - 1/d$.*

**Experimental Results.** We illustrate the practicality of our fingerprinting algorithms by running experiments on two different datasets. In a direct comparison with the implementations of Gulyás et al. (2016), our algorithms show significantly improved efficiency while retaining high comparative accuracy. Specifically we achieve a speedup of up to $49x$ and $210x$ while achieving high comparative accuracy for targeted and generalized fingerprinting, respectively.

We also show that our general fingerprinting algorithm can serve as a dimensionality reduction technique and apply it in the context of feature selection for machine learning models. Since the time complexity of many clustering algorithms including $k$-means scales with the dimensionality of a dataset, we used our algorithm to select $x$ features that best separate the data. Then we ran $k$-means on these $x$ features instead of the full feature space, showing that we increase efficiency while sacrificing little in accuracy. We believe our techniques to be general and extendable to other clustering and machine learning algorithms outside of $k$-means.

## 2. Preliminaries

**Notation.** Some preliminaries are postponed to Appendix A. We denote $A_{ij}$ as the entry at the $i^{\text{th}}$ row and $j^{\text{th}}$ column of matrix $\boldsymbol{A}$. $\tilde{O}(\cdot)$ notation suppresses logarithmic factors in its argument. In general we boldface vectors and matrices.

**Linear Sketches.** We begin by defining what a linear sketch is and then provide an overview of the specific linear sketches used in this paper. A more thorough overview is deferred to Appendix A.1. Given a $n \times d$ matrix $\boldsymbol{A}$, we can compress it while retaining essential information to solve the problem by multiplying it with a $r \times n$ linear sketching matrix $\boldsymbol{S}$. Linear sketches support insertions and deletions to the entries of $\boldsymbol{A}$ as $\boldsymbol{S}(\boldsymbol{A} + c_{ij}) = \boldsymbol{S}\boldsymbol{A} + \boldsymbol{S}c_{ij}$ holds for any update $c_{ij}$. This allows us to maintain $\boldsymbol{S}\boldsymbol{A}$ throughout the updates without storing $\boldsymbol{A}$ itself. Furthermore, $\boldsymbol{S}$ is typically stored in an implicit, pseudorandom form (e.g., via hash functions) rather than explicitly, enabling efficient sketching of updates $c_{ij}$. The update time is the time complexity required for the sketch to process an update.

**CountSketch.** Next we review the Count-Sketch algorithm for frequency estimation (Charikar et al., 2004). Consider an underlying $n$-dimensional vector $\boldsymbol{x}$. The algorithm maintains a $q \times B$ matrix $\boldsymbol{C}$ where $B$ is the number of buckets

---

[2] For the definition of linearly sketchable see Appendix A.3.1.
[3] See Section 2 for more about linear sketches.

or the sketching dimension. It keeps $q$ distinct hash functions $h_i : [n] \to [B]$ and $q$ sign functions $g_i : [n] \to \{-1, 1\}$. The algorithm maintains $C$ such that $C[\ell, b] = \sum_{j:h_\ell(j)=b} g_\ell(j) \cdot x_j$. The frequency estimation $\hat{x}_i$ of $x_i$ is defined to be the median of $\{g_\ell(i) \cdot C[\ell, h_\ell(i)]\}_{\ell \leq q}$. Formally, when $q = O(\log n)$, we have with probability at least $1 - 1/\text{poly}(n)$, $|\hat{x}_i - x_i| \leq O\left(\frac{\|\boldsymbol{x}_{-B/4}\|_2}{\sqrt{B}}\right)$ simultaneously for all $i \in [n]$. Here $\boldsymbol{x}_{-B/4}$ denotes vector $\boldsymbol{x}$ with its top $B/4$ entries (by magnitude) zeroed out. If $\boldsymbol{x}$ has only $B/4$ nonzero entries, with a Countsketch using space $\tilde{O}(B)$ we can recover all $B/4$ entries/frequencies exactly. The update time is $O(\log n)$.

$L_0$ **Sketch.** Consider an underlying vector $\boldsymbol{x} = (x_1, \ldots, x_n)$ where all entries are initially set to 0. We receive $m$ updates of the form $(i, v) \in [n] \times \{-M, ..., M\}$ in a stream where the update performs $x_i \leftarrow x_i + v$. At the end of the stream, the goal is to output a $(1 \pm \varepsilon)$ relative approximation of $L_0$ with probability at least $1 - \delta$ where $L_0 = |\{i : x_i \neq 0\}|$. Kane et al. (2010) give a $L_0$ sketch with $O(1)$ update time that uses $O(\varepsilon^{-2} \log n (\log(1/\varepsilon) + \log \log(mM)) \cdot \log(1/\delta))$ memory.

**Moment Estimation.** Consider an underlying vector $\boldsymbol{x} = (x_1, \ldots, x_n)$. For all $i \in [n]$, $x_i \in [m]$. Let $f_i = |\{j : x_j = i\}|$ be the number of occurrences of value $i$ in $\boldsymbol{x}$. We define the $p^{\text{th}}$ frequency moment of $\boldsymbol{x}$ as $F_p \stackrel{\text{def}}{=} \sum_{i=1}^m f_i^p$ for $p \geq 0$.

## 3. Max-Coverage Algorithm

We now present our sketch, Max-Coverage-LS (Algorithm 5), to prove Theorem 1.1. The proofs are deferred to Appendix B. Recall that the input is formalized as a $n \times d$ matrix $\boldsymbol{A}$, where entry $A_{ij}$ is nonzero if $i$ is in subset $j$, and 0 otherwise. Our approach uses Algorithm 1 from Bateni et al. (2017) as a starting point. Bateni et al. (2017) reduces the original input matrix $\boldsymbol{A}$ to a smaller universe $\boldsymbol{A}_*$ by carefully sampling a subset of its nonzero entries. They then show that running the classical greedy algorithm on this smaller universe yields a $(1 - 1/e - \varepsilon)$-relative approximation for the maximum coverage problem on $\boldsymbol{A}$.

The plan for this section is as follows. We will first introduce the smaller universe $\boldsymbol{A}_*$, describing its properties and role in the problem. Next we will show how we construct $\boldsymbol{A}_*$ under the assumption that the entire input matrix $\boldsymbol{A}$ in its final state is fully accessible. The construction is done in a careful way so that it will be easy to turn it into a linear sketch. Finally, we will remove the assumption that $\boldsymbol{A}$ is fully accessible giving us our final algorithm which will efficiently handle updates.

In the following, note that rows of $\boldsymbol{A}$ and items of $\boldsymbol{A}$ refer to the same thing. Constructing $\boldsymbol{A}_*$ involves permuting the

rows of $\boldsymbol{A}$ and processing them in the order determined by the permutation. For each row $i$ in $\boldsymbol{A}$, a subset of $\tilde{O}(d/(\varepsilon k))$ nonzero entries is arbitrarily selected and added to $\boldsymbol{A}_*$. This process continues until $\boldsymbol{A}_*$ contains $\tilde{O}(d/\varepsilon^3)$ nonzero entries in total. So, $\boldsymbol{A}_*$ is a carefully subsampled version of $\boldsymbol{A}$, where only $\tilde{O}(d/\varepsilon^3)$ of the nonzero entries are retained while the rest are set to 0. We restate their algorithm $\boldsymbol{A}_*(k, \varepsilon, \delta)$ (Algorithm 3) in Appendix B. In Bateni et al. (2017) this subsampled matrix is referred to as $H_{\leq d}$.

Bateni et al. (2017) proves that solving the maximum coverage problem on $\boldsymbol{A}_*$ with a $\alpha$-relative approximation guarantees a $(\alpha - \varepsilon)$-relative approximation on the original matrix $\boldsymbol{A}$ with high probability. The final $(1 - 1/e - \varepsilon)$-relative approximation is achieved using k-cover (Algorithm 4), which sets appropriate parameters and applies the greedy algorithm (or any $(1 - 1/e)$ approximation algorithm) to $\boldsymbol{A}_*$.

**Theorem 3.1** (Theorem 2.7 and 3.1 of Bateni et al. (2017)). *Running k-cover with $\boldsymbol{A}_*$ produces a $(1 - 1/e - \varepsilon)$ approximate solution to maximum coverage with probability $1 - 1/d$.*

We now present building-$\boldsymbol{A}_*$ (Algorithm 1) where we assume we are given complete access to $\boldsymbol{A}$ in its final state along with linear space. Afterwards, we will show how to turn this into a linear sketch which will allow us to efficiently handle updates in sublinear space.

We now prove that building-$\boldsymbol{A}_*$ correctly builds $\boldsymbol{A}_*$ with high probability. Recall that we want $\tilde{O}(d/(\varepsilon k))$ nonzero entries per row until we reach $\tilde{O}(d/\varepsilon^3)$ of them. At a high level, our proof goes as follows. In line 3 we subsample down to a smaller universe $\boldsymbol{A}'$ which only causes us to lose an $\varepsilon$ factor in our approximation. Now in this smaller universe, in line 7 we hash the rows of $\boldsymbol{A}'$ to a bunch of buckets. In lines $8 - 11$, from each bucket, we keep a capped number of nonzero entries and store them. Then starting at line 13, we use all our stored nonzero entries to create $\boldsymbol{A}_*$. We will show that the rows of $\boldsymbol{A}'$ are sufficiently spread out among the buckets so that for each row in $\boldsymbol{A}'$ which has nonzero entries, we keep $\tilde{O}(d/(\varepsilon k))$ of its nonzero entries. We now present our formal proof.

**Lemma 3.2.** *Obtaining an $(1 - 1/e)$ approximate solution to maximum coverage on $\boldsymbol{A}'$ is an $(1 - 1/e - \varepsilon/4)$ approximation solution on $\boldsymbol{A}$ with probability at least $1 - 1/\text{poly}(d)$.*

We denote rows of $\boldsymbol{A}'$ that have at least $d/k$ nonzero entries as "large" and the others as "small". We argue that the number of large items and the total number of nonzero entries among small items is bounded appropriately.

**Lemma 3.3.** *There are at most $O(k \log d/\varepsilon^2)$ large items in $\boldsymbol{A}'$.*

**Lemma 3.4.** *There are $O(\frac{d \log d}{\varepsilon^2})$ total nonzero entries among small items in $\boldsymbol{A}'$.*

---

**Algorithm 1** building-$\boldsymbol{A}_*$ ($n \times d$ matrix $\boldsymbol{A}$, $\varepsilon \in (0,1)$, $k$)

1: Set $\delta = (2 + \log d) \log \log_{1-\varepsilon} n$.
2: Set $\varepsilon = \varepsilon/8$.
3: Subsample rows from $\boldsymbol{A}$ to get $\boldsymbol{A}'$ such that **OPT** in $\boldsymbol{A}'$ is $O(k \log d/\varepsilon^2)$. For clarity, row $j$ in $\boldsymbol{A}'$ and $\boldsymbol{A}$ both correspond to the row vector that corresponds to item $j$.
4: Set $b = O(\frac{k \log d}{\varepsilon^2})$.
5: Set $t = O(\log \frac{d}{\varepsilon})$.
6: **for** $i = 1, \ldots, t$ **do**
7:     Use a hash function to hash each row of $\boldsymbol{A}'$ to $b$ buckets in structure $\mathcal{C}_i$.
8:     **for** each bucket in $\mathcal{C}_i$ **do**
9:         If there are $r$ rows hashed to the bucket, denote the $r$ rows concatenated into a vector of length $rd$ as $\boldsymbol{v}$.
10:         Randomly sample $O(\frac{d \log(1/\varepsilon)}{\varepsilon k})$ nonzero entries from $\boldsymbol{v}$ and store it in $\boldsymbol{A}'_i$.
11:     **end for**
12: **end for**
13: Initialize $\boldsymbol{A}_*(k, \varepsilon)$ as a $n \times d$ matrix with all entries initially set to 0.
14: Let $\mathcal{P}$ be a random permutation of the rows that are in $\boldsymbol{A}'$.
15: **while** the number of nonzero entries in $\boldsymbol{A}_*(k, \varepsilon)$ is less than $\frac{24 d \delta' \log(1/\varepsilon) \log d}{(1-\varepsilon)\varepsilon^3}$ **do**
16:     Process the row $j$ that comes next in $\mathcal{P}$.
17:     Determine among all $i \in [t]$ which $\boldsymbol{A}'_i$ has the most nonzero entries from row $j$. Take this $i$ to be $z$.
18:     **if** row $j$ has less than $\frac{d \log(1/\varepsilon)}{\varepsilon k}$ nonzero entries in $\boldsymbol{A}'_z$ **then**
19:         Add all of the nonzero entries from row $j$ in $\boldsymbol{A}'_z$ to $\boldsymbol{A}_*(k, \varepsilon)$.
20:     **else**
21:         Add $\frac{d \log(1/\varepsilon)}{\varepsilon k}$ of the nonzero entries from row $j$ in $\boldsymbol{A}'_z$, chosen arbitrarily, to $\boldsymbol{A}_*(k, \varepsilon)$.
22:     **end if**
23: **end while**

---

We want to show that for each large item, we recover $d \log(1/\varepsilon)/(\varepsilon k)$ of their nonzero entries from $\boldsymbol{A}'$. In addition, we want to show that for each small item, we recover all their nonzero entries from $\boldsymbol{A}'$. We refer to any item corresponding to a row in $\boldsymbol{A}'$ that contains nonzero entries as a "nonzero" item.

We begin by proving that each nonzero item is hashed to a bucket with no other large item and not too many nonzeros from small items with high probability.

**Lemma 3.5.** *Every nonzero item for some $i \in [t]$ is hashed to a bucket containing*

1. *no other large item*

2. *at most $O(\frac{d \log(1/\varepsilon)}{\varepsilon k})$ nonzero entries from small items*

*with probability $1 - 1/\operatorname{poly}(d)$.*

So by Lemma 3.5, we recover all nonzero entries from small items and $d \log(1/\varepsilon)/(\varepsilon k)$ nonzero entries from each large item present in $\boldsymbol{A}'$ with probability $1 - 1/\operatorname{poly}(d)$. Therefore by combining this with Lemma 3.2, we conclude the proof of correctness for building-$\boldsymbol{A}_*$.

Our final step is to remove the assumption that we have direct access to $\boldsymbol{A}$. We now show how to implement building-$\boldsymbol{A}_*$ via a linear sketch, Max-Coverage-LS (Algorithm 5).

Again recall that we must build $\boldsymbol{A}_*$ while receiving updates to the entries of underlying matrix $\boldsymbol{A}$. We first describe our algorithm and give a high level proof review. Then we will present our formal proof. We defer the pseudocode due to length to Appendix B.

In line 3 of Max-Coverage-LS (Algorithm 5), we first keep a $L_0$ sketch for each column of $\boldsymbol{A}$. This will be useful for the final answer our overall algorithm will output for maximum coverage. Then in line 6, we use a hash function to subsample rows from $\boldsymbol{A}$ to form $\boldsymbol{A}'_m$ for $m \in [\log n]$. Recall that we want to subsample down from $\boldsymbol{A}$ to a smaller universe $\boldsymbol{A}'$ such that **OPT** in $\boldsymbol{A}'$ is $O(k \log d/\varepsilon^2)$. However, we do not know what this sampling rate is since it depends on the contents of $\boldsymbol{A}$. Therefore, we subsample at $\log n$ different levels and form $\log n$ different $\boldsymbol{A}_*$. We will later show how to pick which $\boldsymbol{A}_*$ we will use.

Now for each subsampling rate $m$, we consider the subsampled matrix $\boldsymbol{A}'_m$. Note that we do not store $\boldsymbol{A}'_m$ explicitly since this would not fit in our space allotment. Instead, in this parallel run we only consider updates that affect $\boldsymbol{A}'_m$.

In line 11, we hash the rows of $\boldsymbol{A}'_m$ to $b$ buckets. Note that we are doing this for $t$ iterations to amplify the probability of success. Then for each bucket we consider the rows hashed there as one long vector $\boldsymbol{v}$. Again we do not explicitly store this vector - rather the structures in this bucket only consider

updates relevant to this vector.

Recall that in each bucket our goal is to recover $x = O(d\log(1/\varepsilon)/(\varepsilon k))$ nonzero entries from vector $\boldsymbol{v}$. So, in each bucket in lines $13 - 19$, we subsample the vector $\boldsymbol{v}$ in $\log(rd)$ levels. Then in each level $q$, we keep an $L_0$ sketch and CountSketch structure with $x$ buckets for $\boldsymbol{v}_q$. As stated in Section 2, when keeping a CountSketch structure with $4x$ buckets of a vector that contains at most $x$ nonzero entries, the CountSketch will recover those entries exactly. So we want to identify the sampling level $q'$ where $\boldsymbol{v}_q$ has at most $x$ entries and query the corresponding CountSketch to recover the appropriate number of nonzero entries from $\boldsymbol{v}$. Upon a query, we are identifying the appropriate $q'$ in line 29. Since an $L_0$ sketch returns how many nonzero entries there are in $\boldsymbol{v}_q$ for some $q$, we are simply using the $L_0$ sketches to find $q'$. The rest of the algorithm builds $\boldsymbol{A}_{m,*}$ for $m \in [\log n]$ with our recovered nonzero entries.

We note that in line 25 we state that the $L_0$ sketches and CountSketch structures will handle updates. Recall that $L_0$ sketches and CountSketch structures are both linear sketches. So, they are able to handle insertions and deletions to the vector they are considering. So when an update comes, these linear sketches will update thereby updating our entire linear sketch.

Once we have an $L_0$ sketch for each column of $\boldsymbol{A}$ and all the $\boldsymbol{A}_{m,*}$ for $m \in [\log n]$, we perform the following process, max-coverage (Algorithm 2) to get the final answer.

---

**Algorithm 2** max-coverage

**Require:** $k$ and $\varepsilon \in [0, 1]$.
**Ensure:** A $1 - 1/e - \varepsilon$ approximate solution to maximum coverage with probability $1 - 1/d$.
 1: Set $\varepsilon' = \varepsilon/48$.
 2: For $m \in [\log n]$, construct $\boldsymbol{A}_{m,*}(k, \varepsilon')$ using Max-Coverage-LS (Algorithm 5). Also store the $L_0$ sketches of the columns of $\boldsymbol{A}$ outputted by Algorithm 5.
 3: Run the greedy algorithm (or any $1 - 1/e$ approximation algorithm) on each $\boldsymbol{A}_{m,*}(k, \varepsilon')$.
 4: Use the $L_0$ sketches to determine for which $\boldsymbol{A}_{m,*}$ the greedy algorithm gave the best answer and output it.

---

max-coverage (Algorithm 2) is almost the same as k-cover (Algorithm 4). The difference is we run the classical greedy algorithm on each $\boldsymbol{A}_{m,*}$ for $m \in [\log n]$. Now we need to figure out which $m$ corresponded to the subsampling rate such that **OPT** in $\boldsymbol{A}_{m,*}$ is $O(k\log d/\varepsilon^2)$. Instead of doing this, we just use the answer from the $\boldsymbol{A}_{m,*}$ that will give us the best answer.

Let us say for some $m$ that running the greedy algorithm on $\boldsymbol{A}_{m,*}$ outputs subsets $s_1, \ldots, s_k$. We use the $L_0$ sketches corresponding to $s_1, \ldots, s_k$ to estimate how many distinct

items are covered by those subsets. This allows us to form our final output. Now we present our formal proof.

**Lemma 3.6.** *Max-Coverage-LS (Algorithm 5) and max-coverage (Algorithm 2) correctly implement building-$\boldsymbol{A}_*$ (Algorithm 1) and k-cover (Algorithm 4) with probability at least $1 - 1/\operatorname{poly}(d)$.*

**Lemma 3.7.** *Maximum-Coverage-LS uses $\tilde{O}(d/\varepsilon^3)$ bits of memory.*

**Lemma 3.8.** *The update time of Maximum-Coverage-LS is $\tilde{O}(1)$.*

Note that we incur only a $\varepsilon$ factor loss in total, resulting in a final $1 - 1/e - \varepsilon$ approximation. Specifically, we lose a $\varepsilon/4$ factor going from $\boldsymbol{A}$ to $\boldsymbol{A}'$, another $\varepsilon/4$ factor from running the greedy algorithm on $\boldsymbol{A}_*$, and a $\varepsilon/4$ factor from using the $L_0$ sketches to determine which set of outputs to return. With Lemma 3.6, Lemma 3.7, and Lemma 3.8, we can now conclude the proof of Theorem 1.1.

### 3.1. Targeted Fingerprinting

We now present our proof for Corollary 1.2 via a reduction to maximum coverage. Recall that the input to targeted fingerprinting is $n \times d$ matrix $\boldsymbol{A}$ and target user $u \in [n]$. The proof details are deferred to Appendix B.1.

**Lemma 3.9.** *Take $\boldsymbol{A}'$ to be $\boldsymbol{A}$ with the updates $A_{ij} = A_{ij} - A_{uj}$ applied for all $i \in [n], j \in [d]$. For any union of subsets $\mathcal{U}$, the number of items covered on $\boldsymbol{A}'$ is equivalent to the number of users separated from the target on $\boldsymbol{A}$.*

Algorithmically, we simply run Maximum-Coverage-LS and alongside store the row that corresponds to target user $u$ in $O(d)$ space. After the completion of the updates, we can simulate forming $\boldsymbol{A}'$ from $\boldsymbol{A}$ by sending updates to the maximum coverage sketch for $\boldsymbol{A}$. Therefore, the approximation factor, space, and update time all follow from Theorem 1.1 giving us Corollary 1.2.

### 4. $n^p - F_p$ for $p \geq 2$

We prove Theorem 1.4 with p-Tuples-Sketch (Algorithm 7). Since we know $n$ and $p$, our main goal is to estimate $F_p$. However, existing work which estimates $F_p$ has the error guarantee on $F_p$. In our problem, we want the error guarantee on the complement of $F_p$ (which is $n^p - F_p$). We give a high level algorithm and proof overview. The pseudocode and formal proof are deferred to Appendix D.

The structures we keep are $L_0$ sketches and a number of perfect $\ell_0$ samplers. Recall that an $L_0$ sketch of vector $\boldsymbol{x}$ returns the number of nonzero entries of $\boldsymbol{x}$ (with relative $(1 \pm \gamma)$ error) and an $\ell_0$ sampler of $\boldsymbol{x}$ returns a nonzero entry exactly uniformly at random. Then, upon an update, since $L_0$ sketches and perfect $\ell_0$ samplers are linear sketches, they

can handle arbitrary insertions/deletions to the underlying vector $x$.

Upon a query, the algorithm does the following to output its final approximation to $n^p - F_p$. First it figures out which value has the highest frequency and estimates this frequency using a $L_0$ sketch and the $\ell_0$ samplers. Querying the $L_0$ sketch and subtracting the result from $n$ gives us the number of 0's. The frequencies of the rest of the values can be estimated by looking at their relative frequency among the $\ell_0$ samplers (which can be viewed as a uniform sample of the entries of $x$) and scaling it.

Let us call the highest frequency value $b$ and its corresponding frequency $f_b$. If $f_b > n/2$, then we will estimate the final approximation $f_b$ differently than the rest of the values. Specifically, we subtract off value $b$ from a $L_0$ sketch and then query the result. This gives us the number of entries in $x$ that were not equal to value $b$. Then we set $f'_b$ to be $n$ minus that result. This is important because it gives us an estimate to $f_b$ with $\gamma \cdot (n - f_b)$ error instead of $\gamma \cdot f_b$ error.

Now for the rest of the frequencies, we will again use the $\ell_0$ samplers. We simulate updates that subtract off value $b$ from the underlying vector that the $\ell_0$ samplers consider. This means that the $\ell_0$ samplers are taking uniform samples of $x$ with $b$ subtracted off all the entries. So when estimating the frequencies of the values besides $b$ using the $\ell_0$ samplers after these updates, we have their frequencies with error at most $\gamma \cdot (n - f_b)$ instead of additive error $\gamma n$. For values with very small frequency, we ignore them and show that this does not result in too much error.

## 5. Experiments

We outline our fingerprinting results and compare the runtime/accuracy to Gulyás et al. (2016) [4]. We then present our results on dimensionality reduction. All experiments were run locally on a M2 MacBook Air. The code can be found here. We use two publicly-available datasets, the UC Irvine "Adult" and "US Census Data (1990)" (Becker & Kohavi, 1996; Meek et al.). For consistency, we apply the pre-processing from Gulyás et al. (2016) to both datasets. "Adult" has $32,561$ instances (representing users) and $80$ features while "US Census Data (1990)" has $2,458,285$ instances and $195$ features. We note that for fingerprinting we do not simulate updates since the algorithms of Gulyás et al. (2016) are not streaming algorithms and therefore unable to accommodate updates. We also note that we expect the baseline to always achieve better error. This is because we theoretically lose a small $\varepsilon$ factor in our approximation.

[4](Gulyás et al., 2016) has two implementations, one of which is supposed to be optimized for time. However, we found that the non-optimized implementation was faster and therefore use it for comparison.

However, our experiments show that our algorithms still retain good comparative accuracy and greatly increase the time efficiency.

**Targeted Fingerprinting Results.** We made standard modifications that are done in the practical implementation of streaming algorithms. In particular, we use a constant subsampling rate $p \in [0.1, 0.6]$ instead of subsampling at $\log n$ rates, and we sample nonzero entries once we are in the smaller subsampled universe with a fixed probability as this is sufficient for smaller datasets. All presented data are averages over 10 runs.

We present our results for "Adult" with the probability of subsampling rows from $A$ to create $A'$ to be $p \in [0.1, 0.6]$. One run finds the targeted fingerprint of all users in the dataset for some given $k$. For $k = 7$, from Figure 1 we can

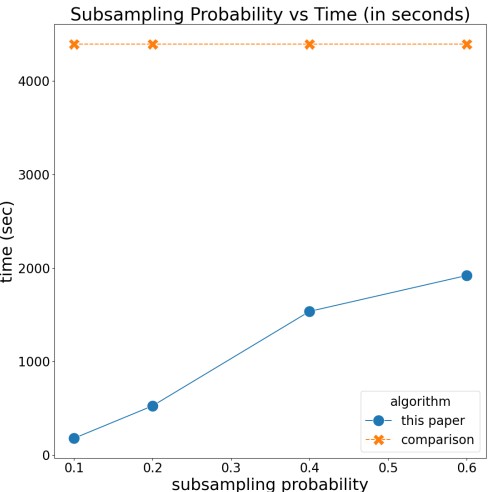

*Figure 1.* "Adult": subsampling probability vs. time

see that our algorithm runs about 25x, 8.4x, 3x, and 2.3x faster than that of Gulyás et al. (2016) with subsampling probabilities 0.1, 0.2, 0.4, and 0.6 respectively. In settings where $n$ is very large the subsampling probability in our algorithm will be much smaller, and we only use larger subsampling probabilities for further insight. Note that the implementation of Gulyás et al. (2016) is deterministic. We put their runtime as a line for visualization. Now we look at accuracy. For increasing $k$, we compute the average percent of users our algorithm is able to separate from a given target user and compare it to (Gulyás et al., 2016). In Figure 2, we show that we retain good accuracy despite subsampling rows and then subsampling nonzero entries. Note that the vertical axis's minimum value is $84\%$. As the subsampling probability increases, the accuracy of our implementation converges to that of Gulyás et al. (2016).

Now, we present our results for "US Census Data (1990)".

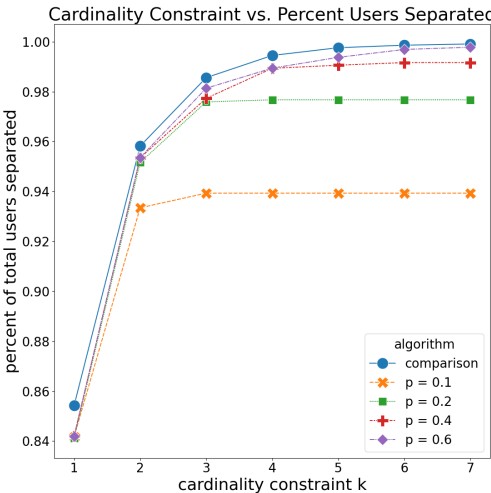

*Figure 2.* "Adult": $k$ vs. accuracy

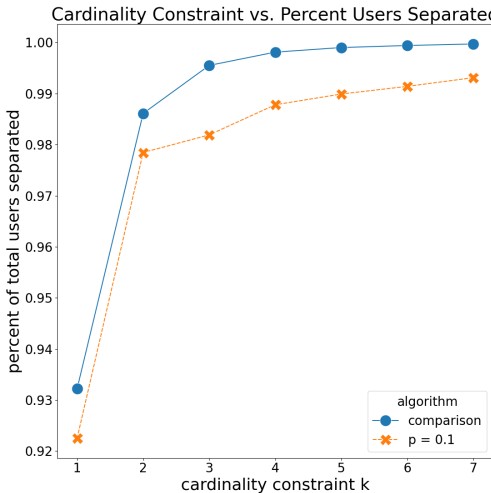

*Figure 3.* "US Census Data": k vs. accuracy

Due to limited compute, we look at one subsampling level of 0.1. Over 10 runs for $k = 7$, the average time of our algorithm to compute a fingerprint for an input user was 1.06 seconds while the comparison average was 52.6 seconds. The subsampling took an extra 46.355 seconds. So, our algorithm was about 49x times faster. For accuracy, Figure 3 shows that we quickly converge to the accuracy of (Gulyás et al., 2016) with growing $k$. Note that the vertical axis's minimum value is 92%.

**General Fingerprinting Results.** The main difference between our theoretical and implemented algorithm is that we only create one sketch rather than $k$ sketches. We first present our results for "Adult". The main variable we vary is the size of our $L_0$ sketch, specifically with $300, 600, 900$, and $1,250$ rows. We had our algorithm compute a general fingerprint for $k = 1, 2, \ldots, 20$. The runtime of our algorithm was largely independent of the size of the sketch and ran in about 0.8 seconds which is 44x faster than the 35.30 second runtime of Gulyás et al. (2016). We measure accuracy by looking at the proportion between the number of pairs of users that our algorithm separates to the number of pairs of users that Gulyás et al. (2016) separates. For each sketch size, we never dip below an accuracy ratio of 80%, and as the sketch size increases the accuracy ratio increases to around 99%. We now present our results for "US Census Data (1990)". We again vary the size of our $L_0$ sketch, using $55,000, 180,000$, and $400,000$ rows. We computed a general fingerprint for $k = 1, 2, \ldots, 10$. We use smaller $k$ for comparison for this dataset since the implementation of Gulyás et al. (2016) was not able to terminate even after several hours for larger $k$.

In Figure 4 we can see that the runtime of our algorithm

increases as the sketch size increases. Our implementation is about $210, 120$, and $45$ times faster than that of Gulyás et al. (2016) for $55,000, 180,000$, and $400,000$ rows respectively. For a fingerprint of size 20 our implementation takes a little over twice the amount of time as for a fingerprint of size 10. We estimate that the runtime of the comparison algorithm also doubles but cannot be sure due to its non-termination. Concerning accuracy, we again see in Figure 5 that as sketch size increases, the accuracy ratio increases. We make note of a steep drop-off for a sketch with $55,000$ rows. However, our accuracy ratio never dips below 70%.

**Dimensionality Reduction Results.** We use the UCI "Wine" dataset which consists of 178 instances and 13 features (Aeberhard & Forina, 1991). Each of the instances is labeled by one of three wine types. We used our general fingerprinting algorithm to select features that best separate the data. Then, we ran $k$-means with 3 clusters (for the 3 wine types) using just the selected features. Therefore, this is a dimensionality reduction technique, since for many clustering algorithms (including $k$-means and $k$-means++) the efficiency depends on the feature dimension. We measure accuracy in the following way. After running $k$-means on the reduced feature space, for each cluster, we calculate the majority wine type. Then, for each instance, if its actual wine type is not the same as the majority wine type of its assigned cluster, we count it towards the error. We used general fingerprinting to reduce the feature dimension to $3, 4$, and $5$ features. Our accuracy for all was around 68%. When running $k$-means using all 12 features, the accuracy was around 71%, which suggests that we do not introduce that much error. In addition, when running $k$-means instead on just $3, 4$, and $5$ completely randomly chosen features, the accuracy decreases to around 52%. We also increase

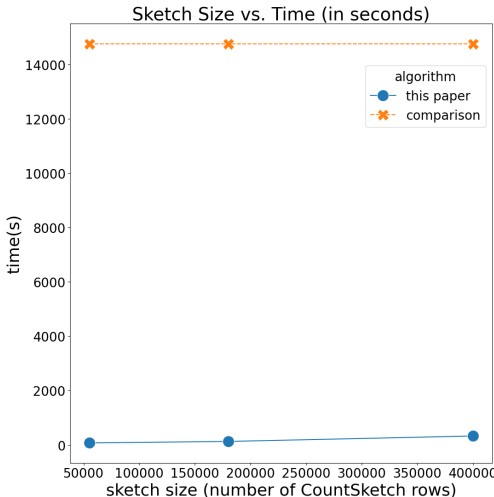

Figure 4. "US Census Data": sketch size vs. time

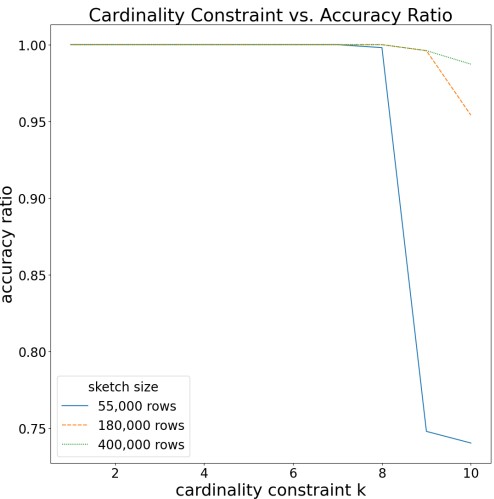

Figure 5. "US Census Data": k vs. accuracy

the efficiency of running $k$-means. Running $k$-means with our reduced $3, 4,$ and $5$ features compared to running it with all $13$ features is about $3.2, 2.4,$ and $2.1$ times faster, respectively.

## Impact Statement

This paper presents work whose goal is to advance the field of Machine Learning. There are many potential societal consequences of our work, none which we feel must be specifically highlighted here.

## Acknowledgments

Hoai-An Nguyen was supported in part by an NSF GRFP fellowship grant number DGE2140739 and NSF CAREER Award CCF-2330255. Huy Nguyen was supported in part by NSF award number 2311649. Alina Ene was supported in part by NSF CAREER award CCF-1750333 and an Alfred P. Sloan Research Fellowship.

We thank Sepehr Assadi for numerous helpful discussions. We thank William He for giving many useful presentational comments. We also thank Praneeth Kacham, Noah Singer, and Brian Zhang for helping us review the paper and giving useful comments.

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

# A. Extended Preliminaries

### A.1. Linear Sketches

Given a $n \times d$ matrix $\boldsymbol{A}$, we can compress it while retaining essential information to solve the problem by multiplying it with a $r \times n$ linear sketching matrix $\boldsymbol{S}$. A linear sketch is a matrix drawn from a certain family of random matrices independent of $\boldsymbol{A}$. This independence ensures that $\boldsymbol{S}$ can be generated without prior knowledge of the contents of $\boldsymbol{A}$. Linear sketches support insertions and deletions to the entries of $\boldsymbol{A}$, as $\boldsymbol{S}(\boldsymbol{A} + c_{ij}) = \boldsymbol{S}\boldsymbol{A} + \boldsymbol{S}c_{ij}$ holds for any update $c_{ij}$, which adds or subtracts $c$ from an entry of $\boldsymbol{A}$. This allows us to maintain $\boldsymbol{S}\boldsymbol{A}$ throughout the updates without storing $\boldsymbol{A}$ itself. Furthermore, $\boldsymbol{S}$ is typically stored in an implicit, pseudorandom form (e.g., via hash functions) rather than explicitly, enabling efficient sketching of updates $c_{ij}$. The primary focus is on minimizing the space requirement of a linear sketch, specifically ensuring that the sketching dimension $r$ is sublinear in $n$ and ideally much smaller. Another important metric is the update time. Update time refers to the time complexity required for the sketch to process an update.

### A.2. Perfect $\ell_0$ Sampling.

Consider an underlying vector $\boldsymbol{x} = (x_1, \ldots, x_n)$. Let $\text{Supp}(\boldsymbol{x})$ be the set of nonzero elements of $\boldsymbol{x}$. A perfect $\ell_0$ sampler, with probability $1 - \delta$, returns a tuple $(i, x_i)$ for $x_i \in \text{Supp}(\boldsymbol{x})$ such that $\Pr[i = j] = \frac{1}{\|\boldsymbol{x}\|_0} \pm n^{-c}$ for every $j$ such that $x_j \in \text{Supp}(\boldsymbol{x})$ for large constant $c$. Note that it returns the value of $x_i$ exactly with no error. With probability $\delta$, the sampler outputs FAIL.

An $\ell_0$ sampler is a linear sketch and accommodates both insertions and deletions to the underlying vector $\boldsymbol{x}$. The parameter $n^{-c}$ can be made arbitrarily small by increasing constant $c$, effectively making the sampling process indistinguishable from perfect uniform random sampling of nonzero entries on polynomial length streams. Importantly, increasing $c$ incurs only constant factors in space usage. Jowhari et al. (2011) give a $\ell_0$ sampler that uses $O(\log^2 n \log(1/\delta))$ bits of space. By inspecting Theorem 2 of Jowhari et al. (2011) and using appropriate sparse recovery schemes we can see that the update time is $\text{poly}(\log n) \cdot \log(1/\delta)$.

### A.3. Streaming Model

In this paper, we represent the input as a $n \times d$ matrix $\boldsymbol{A}$. In the streaming model, it is standard to initialize all the entries to zero before the stream of updates. The algorithm then processes a stream of updates which come one-by-one, each of the form $(i, j, \pm c)$ for some $c$. This modifies entry $A_{ij}$ by performing $A_{ij} = A_{ij} + c$ or $A_{ij} = A_{ij} - c$ depending on the sign. We make the standard assumption that each $c$ is at most $\text{poly}(n)$.

When the updates can only be positive, this is the insertion-only streaming model. When updates can be both positive and negative this is referred to as the turnstile streaming model. The updates can appear in any arbitrary order in the stream, and we make the standard assumption that the length of the stream is at most $\text{poly}(n)$. The goal of the streaming algorithm is to process the stream efficiently, using sublinear space in the size of the input matrix $\boldsymbol{A}$ (and therefore it cannot store all the updates) and a small constant number of passes over the stream. In this work, we restrict our focus to one-pass algorithms. At the end of the stream, the algorithm can do some post-processing and then must output the answer. While streaming algorithms are not required to maintain a stored answer at every point during the stream, there is no restriction on when the stream may terminate. Any time or space used before or after processing the stream is attributed to pre-processing or post-processing, respectively. Generally, our primary focus is on optimizing the memory usage and update time during the stream. Here the update time is the time complexity required by the algorithm to process an update.

#### A.3.1. Useful Definitions

**Monotone Submodular Maximization.** Consider a non-negative set function $f : 2^V \to \mathbb{R}_+$. If for all $S \subseteq T \subseteq V \setminus \{e\}$, $f$ satisfies: $f(S \cup \{e\}) - f(S) \geq f(T \cup \{e\}) - f(T)$, then $f$ is submodular. We assume that $f(\emptyset) = 0$. If $f(S) \leq f(T)$ for all $S \subseteq T$, then $f$ is also monotone.

**Linearly Sketchable Functions.** All the functions $f : 2^d \to \mathbb{R}_+$ that we consider are of the form $f(\mathcal{C}) = g(\{\boldsymbol{a}_i\}_{i \in \mathcal{C}})$ where $\boldsymbol{a}_1, \ldots, \boldsymbol{a}_d$ are a set of vectors that are either fixed in advance or are the columns of the $n \times d$ matrix $\boldsymbol{A}$ that is being updated in the stream. We say that a function $f$ is "linearly sketchable" if there exists a randomized sketching matrix $\boldsymbol{S}$ and a corresponding function $g_{\boldsymbol{S}}$ such that, for any vectors $\boldsymbol{a}_1, \ldots, \boldsymbol{a}_d$, with high probability for all $\mathcal{C} \subseteq [d]$, $f(\mathcal{C})$ can be approximated by $g_{\boldsymbol{S}}(\{\boldsymbol{S} \cdot \boldsymbol{a}_i\}_{i \in \mathcal{C}})$.

A.3.2. CONCENTRATION INEQUALITIES

**Markov's Inequality.** If $X$ is a nonnegative random variable and $a > 0$, then

$$\Pr(X \geq a) \leq \frac{\mathbb{E}\left[X\right]}{a}.$$

**Chebyshev's Inequality.** For any random variable $X$ and $t > 0$.

$$\Pr(|X - \mathbb{E}\left[X\right]| \geq t) \leq \frac{\mathrm{Var}[X]}{t^2}.$$

# B. Deferred Parts of Section 3 (Maximum Coverage)

We restate the algorithm of Bateni et al. (2017), $\boldsymbol{A}_*(k, \varepsilon, \delta)$ (Algorithm 3).

---

**Algorithm 3** $\boldsymbol{A}_*(k, \varepsilon, \delta)$

---

**Require:** $k, \varepsilon \in (0, 1]$, and $\delta$.
**Ensure:** $\boldsymbol{A}_*(k, \varepsilon, \delta)$.
 1: Let $\delta' = \delta \log \log_{1-\varepsilon} n$.
 2: Let $h$ be an arbitrary hash function that uniformly and independently maps each item (or each row of $\boldsymbol{A}$) to $[0, 1]$.
 3: Initialize $\boldsymbol{A}_*(k, \varepsilon, \delta)$.
 4: **while** number of nonzero entries in $\boldsymbol{A}_*(k, \varepsilon, \delta)$ is less than $\frac{24 d \delta' \log(1/\varepsilon) \log d}{(1-\varepsilon)\varepsilon^3}$ **do**
 5:   Pick item $i$ of minimum $h(i)$ that has not been considered yet.
 6:   **if** there are less than $\frac{d \log(1/\varepsilon)}{\varepsilon k}$ nonzero entries in the $i^{\text{th}}$ row of $\boldsymbol{A}$ **then**
 7:     Add all the nonzero entries from the $i^{\text{th}}$ row of $\boldsymbol{A}$ to $\boldsymbol{A}_*(k, \varepsilon, \delta)$.
 8:   **else**
 9:     Add $\frac{d \log(1/\varepsilon)}{\varepsilon k}$ of the nonzero entries of $\boldsymbol{A}$, chosen arbitrarily, to $\boldsymbol{A}_*(k, \varepsilon, \delta)$.
10:   **end if**
11: **end while**

---

We now restate the final algorithm from Bateni et al. (2017), $k$-cover (Algorithm 4).

---

**Algorithm 4** $k$-cover

---

**Require:** $k$ and $\varepsilon \in [0, 1]$.
**Ensure:** A $(1 - 1/e - \varepsilon)$ approximate solution to maximum coverage with probability $1 - 1/d$.
 1: Set $\delta = 2 + \log d$ and $\varepsilon' = \varepsilon/12$.
 2: Construct $\boldsymbol{A}_*(k, \varepsilon', \delta)$.
 3: Run the greedy algorithm (or any $1 - 1/e$ approximation algorithm) on $\boldsymbol{A}_*(k, \varepsilon', \delta)$ and report the output.

---

**Lemma B.1.** *Obtaining an $(1-1/e)$ approximate solution to maximum coverage on $\boldsymbol{A}'$ is an $(1-1/e-\varepsilon/4)$ approximation solution on $\boldsymbol{A}$ with probability at least $1 - 1/\mathrm{poly}(d)$.*

*Proof.* This states that we only lose a $\varepsilon/4$ factor by reducing to a smaller universe via subsampling such that $\mathbf{OPT} = O(k \log d/\varepsilon^2)$. This is proven in McGregor & Vu (2019) in Corollary 9. Note that in McGregor & Vu (2019) they prove a $(1 - 1/e)$ approximate solution on $\boldsymbol{A}'$ is an $(1 - 1/e - 2\varepsilon)$-approximation solution on $\boldsymbol{A}$ but we re-weigh $\varepsilon$ in our algorithm. □

**Lemma B.2.** *There are at most $O(k \log d/\varepsilon^2)$ large items in $\boldsymbol{A}'$.*

*Proof.* Suppose that there are $\ell$ large items. Take $\mathcal{S}$ to be the set of these large items. Take $\mathcal{P}$ to be the set of the $d$ input subsets. Now we will conduct the following process. First, we choose a subset $p_1 \in \mathcal{P}$. Suppose that $p_1$ covers $c_1$ large items from $\mathcal{S}$. Recall that "covers" means that those items are in subset $p_1$. Now remove $p_1$ from $\mathcal{P}$ and the $c_1$ large items it

covered from $\mathcal{S}$. Now let us choose another subset $p_2 \in \mathcal{P}$. Suppose that $p_2$ covers $c_2$ of the (remaining) items in $\mathcal{S}$. Again remove $p_2$ from $\mathcal{P}$ and the $c_2$ large items it covered from $\mathcal{S}$. Continue this process for a total of $k$ times.

Since we know that $\mathbf{OPT} = C_1 \cdot k \log d / \varepsilon^2$ for some constant $C_1$, it must be that $c_1 + c_2 + \cdots + c_k \leq C_1 \cdot k \log d / \varepsilon^2$. Now, suppose for the sake of contradiction that $\ell = C_2 \cdot k \log d / \varepsilon^2$ for some constant $C_2$. Then,

$$C_2 \cdot k \log d / \varepsilon^2 - c_1 - \cdots - c_k > C_2 \cdot k \log d / \varepsilon^2 - C_1 \cdot k \log d / \varepsilon^2 > C_2 \cdot k \log d / (2\varepsilon^2)$$

for $C_2 > 2C_1$.

So, at each step of the above process, there were at least $C_2 \cdot k \log d / (2\varepsilon^2)$ large items left in $\mathcal{S}$ and therefore at least $C_2 d \log d / (2\varepsilon^2)$ nonzero entries among those large items. So, at the end of the process we could have covered $k \cdot C_2 \log d / (2\varepsilon^2)$ large items. But then we would have

$$\mathbf{OPT} \geq C_2 / 2 \cdot k \log d / \varepsilon^2 > C_1 \cdot k \log d / \varepsilon^2$$

which is a contradiction. $\qquad\square$

**Lemma B.3.** *There are $O(\frac{d \log d}{\varepsilon^2})$ total nonzero entries among small items in $\mathbf{A}'$.*

*Proof.* Take $\mathcal{S}$ to be the set of small items. Take $\mathcal{P}$ to be the set of the $d$ input subsets. Suppose that there are $s$ total nonzero entries among the items in $\mathcal{S}$. Now we will conduct the following process. First we choose a subset $p_1 \in \mathcal{P}$. Suppose that $p_1$ covers $c_1$ small items from $\mathcal{S}$. Recall that "covers" means that those items are in subset $p_1$. Now remove $p_1$ from $\mathcal{P}$ and the $c_1$ small items it covered from $\mathcal{S}$. Now let us choose another subset $p_2 \in \mathcal{P}$. Suppose that $p_2$ covers $c_2$ of the (remaining) items in $\mathcal{S}$. Again remove $p_2$ from $\mathcal{P}$ and the $c_2$ small items it covered from $\mathcal{S}$. Continue this process for a total of $k$ times.

We know that $\mathbf{OPT} = C_1 \cdot k \log d / \varepsilon^2$ for some constant $C_1$. Therefore, we have that in the above process we removed at most

$$(c_1 + \ldots + c_k) \cdot \frac{d}{k} \leq C_1 \cdot \frac{d \log d}{\varepsilon^2}$$

edges since we have $c_1 + \ldots + c_k \leq \mathbf{OPT}$.

Now suppose for the sake of contradiction that $s = C_2 \cdot d \log d / \varepsilon^2$ for some constant $C_2$. But then, in the above process, at each step we could have found a subset covering at least $(C_2 - C_1) \cdot \log d / \varepsilon^2$ new items. This would mean that we have $\mathbf{OPT} \geq k \cdot (C_2 - C_1) \cdot \log d / \varepsilon^2$ which for appropriate $C_2$ is a contradiction.

$\qquad\square$

**Lemma B.4.** *Every nonzero item for some $i \in [t]$ is hashed to a bucket containing*

1. *no other large item*

2. *at most $O(\frac{d \log(1/\varepsilon)}{\varepsilon k})$ nonzero entries from small items*

*with probability $1 - 1/\operatorname{poly}(d)$.*

*Proof.* Consider some nonzero item $x$. Now consider some iteration $i \in [t]$.

We will first consider the large items. By Lemma 3.3, there are at most $C_1 \cdot k \log d / \varepsilon^2$ large items for some constant $C_1$. $\mathcal{C}_i$ has $C_2 \cdot k \log d / \varepsilon^2$ buckets. For appropriate $C_2$, we can say that $C_2 > 4C_1$. In the worst case, every large item is hashed to a different bucket. Then, the probability of $x$ being hashed to a bucket with another large item is at most $1/4$.

Now let us look at the small items. By Lemma 3.4, there are at most $O(d \log d / \varepsilon^2)$ total nonzero entries among small items. Since small items have at most $d/k$ nonzero entries, in the worst case there are $O(k \log d / \varepsilon^2) = C_3 \cdot k \log d / \varepsilon^2$ small items each with $d/k$ edges. Recall that $\mathcal{C}_i$ has $C_2 \cdot k \log d / \varepsilon^2$ buckets. Therefore, the expected number of nonzero entries from small items in the same bucket as $x$ is at most $C_3 d / C_2 k \leq 1/4 \cdot d/k$. By Markov's inequality, the probability that the actual number of nonzero entries from small items in the same bucket as $x$ is more than $d/k$ is at most $1/4$.

So, the probability that $x$ is hashed to a bucket containing another large item or more than $d/k$ nonzero entries from small entries is at most $1/2$.

We have $O(d \log(1/\varepsilon)/(\varepsilon k)) = C_4 \cdot d \log(1/\varepsilon)/(\varepsilon k) \geq d/k$ for appropriate $C_4$ and $\varepsilon \in (0, 1/2)$. However note that our proof still holds for the full range of $\varepsilon$ since we can always use a smaller $\varepsilon$ to achieve the desired error bound while only incurring an extra constant factor in the space/time.

We hash $O(\log(d/\varepsilon))$ times (since we do it for $i = 1, \ldots, t$). Since we have at most $\tilde{O}((k + d)/\varepsilon^2)$ nonzero items by Lemma 3.3 and Lemma 3.4, we have the result by taking a union bound. $\qquad\square$

We show how to implement building-$\boldsymbol{A}_*$ via a linear sketch, Max-Coverage-LS (Algorithm 5).

**Lemma B.5.** *Max-Coverage-LS (Algorithm 5) and max-coverage (Algorithm 2) correctly implement building-$\boldsymbol{A}_*$ (Algorithm 1) and k-cover (Algorithm 4) with probability at least $1 - 1/\operatorname{poly}(d)$.*

*Proof.* The first step in building-$\boldsymbol{A}_*$ is subsampling from $\boldsymbol{A}$ to get $\boldsymbol{A}'$ such that **OPT** in $\boldsymbol{A}'$ is $O(k \log d/\varepsilon^2)$. Since this sampling rate depends on what **OPT** is in $\boldsymbol{A}$, in Max-Coverage-LS, we instead sample in $\log n$ different rates. So in one of the $\log n$ different parallel runs, we will sample with the correct rate. We will describe how we choose the right run to consider later.

Let us consider the parallel run with the correct sampling rate. In each iteration $i \in [t]$, for each bucket, we need to show that we recover $x = O(\frac{d \log(1/\varepsilon)}{\varepsilon k})$ nonzero entries from $\boldsymbol{v}$. The length of $\boldsymbol{v}$ is $rd$. So, we subsample in $\log(rd)$ levels. At each level we keep an $L_0$ sketch python and CountSketch structure with $x$ buckets. At some level $q'$ we will have that the number of nonzero entries of $\boldsymbol{v}_{q'} \leq x/4$. We use the $L_0$ sketch to find $q'$. By the correctness of CountSketch (see Section 2) we will exactly recover the entries of $\boldsymbol{v}_{q'}$. Note that we set the failure probability appropriately for the CountSketch structures and $L_0$ sketches so we only incur $1/\operatorname{poly}(d)$ total error.

So Max-Coverage-LS (Algorithm 5) produces a $L_0$ sketch for each column of $\boldsymbol{A}$ and $\boldsymbol{A}_{m,*}$ for $m \in [\log n]$. We must figure out which $\boldsymbol{A}_{m,*}$ is the one that corresponds to the desired subsampling rate. We instead find which $\boldsymbol{A}_{m,*}$ gives us the best answer on the original input $\boldsymbol{A}$ using the $L_0$ sketches in the following way.

Suppose that for some $\boldsymbol{A}_{m,*}$ the greedy algorithm chooses subsets $s_1, \ldots, s_k$. We take the $L_0$ sketches for these subsets (or columns of $\boldsymbol{A}$) and reduce to the vector case to estimate how many distinct items these subsets cover in their union.

Imagine that we are working with the original input $\boldsymbol{A}$. Now, take the original columns $s_1, \ldots, s_k$ and concatenate them into a $n \times k$ matrix $\boldsymbol{L}$. We now randomly generate a $k \times 1$ vector $\boldsymbol{x}$ with entries between $[-\operatorname{poly}(d), \operatorname{poly}(d)]$. Now multiply $\boldsymbol{L}$ by $\boldsymbol{x}$. We can see with probability at least $1 - 1/\operatorname{poly}(d)$, the $i^{\text{th}}$ entry in $\boldsymbol{L} \cdot \boldsymbol{x}$ is nonzero if and only if the $i^{\text{th}}$ row of $\boldsymbol{L}$ is nonzero. So, if the $i^{\text{th}}$ entry of $\boldsymbol{L} \cdot \boldsymbol{x}$ is nonzero, that means the $i^{\text{th}}$ item was covered by the union of the subsets $s_1, \ldots, s_k$.

Note that $\boldsymbol{L} \cdot \boldsymbol{x}$ is by definition equivalent to summing $\boldsymbol{L}_1 \cdot x_1 + \boldsymbol{L}_2 \cdot x_2 + \cdots + \boldsymbol{L}_k \cdot x_k$ where $\boldsymbol{L}_i$ denotes the $i^{\text{th}}$ column of $\boldsymbol{L}$ and $x_i$ denotes the $i^{\text{th}}$ entry of $\boldsymbol{x}$. Since the $L_0$ sketches are linear sketches, by definition they have the property that the $L_0$ sketch of the sum of two vectors is equivalent to summing the $L_0$ sketches for the two vectors (see Section 2). Therefore, using the $L_0$ sketches for the $d$ columns of $\boldsymbol{A}$, we can create the $L_0$ sketch for $\boldsymbol{L} \cdot \boldsymbol{x}$ and query it to get a $(1 + \varepsilon/4)$ approximation to the true coverage of the union of subsets $s_1, \ldots, s_k$. $\qquad\square$

**Lemma B.6.** *Maximum-Coverage-LS uses $\tilde{O}(d/\varepsilon^3)$ bits of memory.*

*Proof.* Keeping a $L_0$ sketch for each column of $\boldsymbol{A}$ requires $\tilde{O}(d/\varepsilon^2)$ space total.

The other structures we store are the $L_0$ sketches and CountSketch structures in each bucket. We have $\log n$ subsampling instances. In each instance we store $O(b \log(d/\varepsilon))$ buckets for $b = O(k \log d/\varepsilon^2)$. In each bucket we store $O(\log(rd))$ $L_0$ sketches and $O(\log(rd))$ CountSketch structures with sketching dimension (i.e. the number of buckets in the CountSketch structure) $O(d \log(1/\varepsilon)/(\varepsilon k))$. So the total space requirement of the $L_0$ sketches and CountSketch structures is $\tilde{O}(d/\varepsilon^3)$. $\qquad\square$

**Lemma B.7.** *The update time of Maximum-Coverage-LS is $\tilde{O}(1)$.*

*Proof.* We have an $L_0$ sketch for each column of $\boldsymbol{A}$. Each update will only cause one of these to update. So the update time here is $O(1)$.

Let's look at the update time of the $L_0$ sketches in the buckets. Each update will only cause $\tilde{O}(1)$ of them to update. Note that only one bucket gets affected with an update and we only have to additionally consider the multiple subsampling levels and iterating to boost success probability. So the update time here is $\tilde{O}(1)$.

We finally consider the update time of the CountSketch structures in each bucket. Again only one bucket gets affected during an update. So the update time here is also $\tilde{O}(1)$. □

### B.1. Targeted Fingerprinting

**Lemma B.8.** *Take $\mathbf{A}'$ to be $\mathbf{A}$ with the updates $A_{ij} = A_{ij} - A_{uj}$ applied for all $i \in [n], j \in [d]$. For any union of subsets $\mathcal{U}$, the number of items covered on $\mathbf{A}'$ is equivalent to the number of users separated from the target on $\mathbf{A}$.*

*Proof.* For all $i \in [n]$, for any $j \in [d]$ such that $A_{ij} = A_{uj}$, we have $A'_{ij} = 0$. Additionally, for all $i \in [n]$, for any $j \in [d]$ such that $A_{ij} \neq A_{uj}$, we have that $A'_{ij}$ is nonzero.

In other words, for all users, for any feature where they shared the same value with the target user $u$, this entry is now 0. In addition, for any feature where they did not share the same value with the target user, this entry is now nonzero. We can see that the maximum coverage problem on $\mathbf{A}'$ exactly corresponds to finding $k$ features which separates the most users from target user $u$ on $\mathbf{A}$. □

## C. Proof of Theorem 1.3 (Submodular Maximization Framework)

Here, we outline a framework to design algorithms to maximize monotone non-negative submodular functions that are linearly sketchable subject to a cardinality constraint. At a high level we will receive a linear sketch of the input matrix $\mathbf{A}$ such that querying the sketch will produce the function's output value on some union of subsets. We then adapt the classical greedy algorithm for maximizing a monotone submodular function to query the linear sketches instead of accessing the input matrix directly.

We note that setting $\gamma = \varepsilon/k$ for many linear sketches introduces $\text{poly}(k)$ factors in the final memory usage. However, setting $\gamma = \varepsilon/k$ is provably necessary when performing submodular maximization over queried function values that are preserved up to a $(1 \pm \gamma)$ factor to achieve a $1 - 1/e - \varepsilon$ approximation (see Theorem 5 of Horel & Singer (2024)). Note that this applies to all algorithms that perform submodular maximization that have this property.

We now prove Theorem 1.3. Theorem 1.3 allows us to create an algorithm to maximize a *specific* monotone non-negative submodular function subject to a cardinality constraint by simply sketching the input $\mathbf{A}$ via a linear sketch that satisfies the properties of the theorem.

Let $\mathcal{C}$ be a subset of the column vectors of $\mathbf{A}$. In the following, $\{\mathbf{S} \cdot \mathbf{a}_i\}_{i \in \mathcal{C}}$ can be thought of as the sketch of $\mathbf{A}$ restricted to $\mathcal{C}$. As described in Appendix A.3.1, we say that our function $f$ has a corresponding sketching matrix $\mathbf{S}$ and $g_{\mathbf{S}}$. Recall that $g_{\mathbf{S}}$ gives the answer to a query of function $f$ based on $\mathbf{S}$. For any two subsets of columns $X$ and $Y$, let $g_{\mathbf{S}}(\{\mathbf{S} \cdot \mathbf{a}_i\}_{i \in X|Y})$ denote the marginal gain of adding $X$, or $g_{\mathbf{S}}(\{\mathbf{S} \cdot \mathbf{a}_i\}_{i \in X \cup Y}) - g_{\mathbf{S}}(\{\mathbf{S} \cdot \mathbf{a}_i\}_{i \in Y})$. Take $c \in d \setminus \mathcal{C}$ to denote a column $c$ which is not already in subset $\mathcal{C}$.

We now present our algorithm, sketchy-submodular-maximization (Algorithm 6). We first create $k$ independent linear sketches (recall that the process of creating a linear sketch for the input function is given as input to the algorithm). Then we run the following classical greedy submodular maximization algorithm with the modification that instead of directly evaluating the input function $f$ it queries the given sketch. Note that in each of the $k$ adaptive rounds, we use a different sketch. The classical greedy algorithm in each round simply looks at all subsets that have not been chosen and adds the one with the largest marginal gain to the output set (Nemhauser et al., 1978).

We first analyze the memory usage. We are given that each sketch takes $O(s)$ space. Since there are $k$ rounds of adaptivity, the total space taken by the sketches is $O(sk)$. The update time will depend on the specific linear sketch.

Now, let us prove correctness. We assume by our theorem statement that our sketch $\mathbf{S}$ and corresponding function $g_{\mathbf{S}}$ give us a $(1 \pm \gamma)$-approximation to the queried values of our input function $f$. There are $k$ adaptive rounds. Since we create as many sketches and use a different one in each round, adaptivity between the rounds does not introduce error. In addition, despite getting $(1 \pm \gamma)$-approximations to all our queried values instead of the true queried values of our input function, we still get our desired approximation ratio by setting $\gamma = \varepsilon/k$. This is proven and discussed in Theorem 5 of Horel & Singer

(2024).

We also get the desired output with probability at least $1 - 1/\operatorname{poly}(d)$. Since the error probability for each function evaluation is $O(\frac{1}{\operatorname{poly}(d)k})$, by a union bound over all $dk$ function evaluations, we have an error probability of at most $1 - 1/\operatorname{poly}(d)$.

## D. Proof of Theorem 1.4 ($n^p - F_p$ for $p \geq 2$)

We first present the algorithm, p-Tuples-Sketch.

**Lemma D.1.** *p-Tuples-Sketch uses $\tilde{O}(\gamma^{-\frac{2}{p-1}})$ space and has an update time of $\tilde{O}(\gamma^{-\frac{2}{p-1}})$.*

*Proof.* We keep two $L_0$ sketches and $2t = \tilde{O}(\gamma^{-\frac{2}{p-1}})$ perfect $\ell_0$ samplers. Recall that $p$ is a constant. $\square$

Now we prove correctness. We first give the following result which we will use throughout the proof.

**Lemma D.2** (Lemma 3 of Bhattacharyya et al. (2016))**.** *Let $f_i$ and $\hat{f}_i$ be the frequencies of an item $i$ in a stream $\mathcal{S}$ (of length $n$) and in a random sample of $\mathcal{T}$ of size $r$ from $\mathcal{S}$, respectively. Then for $r \geq 2\gamma^{-2}\log(2\delta^{-1})$, with probability $1 - \delta$, for every universe item $i$ simultaneously,*

$$\left| \frac{\hat{f}_i}{r} - \frac{f_i}{n} \right| \leq \gamma.$$

We will assume that all $L_0$ sketches and $\ell_0$ samplers succeed in the following, which is true with probability $1 - 3\delta/8$. By Lemma D.2, we have that using the $\ell_0$ samplers (from $\mathcal{S}_1$ and $\mathcal{S}_2$) to estimate the frequencies incurs error at most $2\delta/8$. So, we show that we incur at most $3\delta/8$ error in the rest of the algorithm.

For the rest of the analysis, let us order the frequencies of the distinct values of vector $x$ in non-increasing order as $f_1 \geq f_2 \geq \ldots \geq f_z$.

In the algorithm we determine the value of highest frequency and estimate its frequency as $f_b'$. If $f_b'$ is too small, we simply output $n^p$. We now show that this is a good approximation.

**Lemma D.3.** *If $f_1 \leq \gamma^{\frac{1}{p-1}} \cdot n$, then $n^p$ is a $(1 \pm \gamma^{\frac{1}{p-1}})$ approximation to $n^p - \sum_{i \in \mathcal{Z}} f_i^p$.*

*Proof.* Since $f_1 \leq \gamma^{\frac{1}{p-1}} \cdot n$, it must be that $f_i \leq \gamma^{\frac{1}{p-1}} \cdot n$ for all $i$. In this case, $\sum_{i \in \mathcal{Z}} f_i^p$ is maximized when there are $1/\gamma^{\frac{1}{p-1}}$ values each with true frequency $\gamma^{\frac{1}{p-1}} \cdot n$. So we have

$$\sum_{i \in \mathcal{Z}} f_i^p \leq \sum_{i=1}^{(\frac{1}{\gamma})^{\frac{1}{p-1}}} (\gamma^{\frac{1}{p-1}} \cdot n)^p = \left(\frac{1}{\gamma}\right)^{\frac{1}{p-1}} \cdot \gamma^{\frac{p}{p-1}} \cdot n^p = \gamma \cdot n^p.$$

$\square$

Recall that in the algorithm we have $\varepsilon = \frac{\gamma^{\frac{1}{p-1}}}{16 \cdot 2^p}$.

**Lemma D.4.** *For all values $v$ simultaneously, $f_v' = f_{\mathcal{S}_1,v}' \cdot w_1/t = f_v \pm 3\varepsilon n$. $f_{\mathcal{S}_1,v}'$ denotes the frequency of $v$ among the uniform samples which make up $\mathcal{S}_1$.*

*Proof.* By the correctness of $L_0^1$, we have that $w_2$ is a $(1 \pm \varepsilon)$ relative approximation to the number of nonzero entries in $x$. By Lemma D.2, we incur at most $\varepsilon n$ additive error for estimating each $f_v'$ simultaneously using $\mathcal{S}_1$ if $w_2$ was the exact number of nonzeros in $x$. So the combined error is at most $(2\varepsilon + \varepsilon^2) \cdot n \leq 3\varepsilon \cdot n$. $\square$

In addition, we use $L_0^1$ to determine the number of 0's to see if 0 is the value of the largest frequency. This incurs at most $\varepsilon \cdot n$ error. Recall that in the algorithm we output $n^p$ if we estimate $f_b'$ to be less than $\frac{3\gamma^{\frac{1}{p-1}}}{4} \cdot n$. Since by Lemma D.4 and the correctness of $L_0^1$ we know the error in estimating each frequency is at most $3\varepsilon \cdot n \leq \frac{\gamma^{\frac{1}{p-1}}}{4} \cdot n$, at worst all values had frequency $\gamma^{\frac{1}{p-1}} \cdot n$, and we output $n^p$. This does not incur too much error by Lemma D.3.

In the rest of the analysis, we can assume that $f_1 \geq \frac{\gamma^{\frac{1}{p-1}}}{2} \cdot n$. We now claim that incurring additive $\gamma^{\frac{1}{p-1}} \cdot f_1^{p-1} \cdot (n - f_1)$ error still gives us the desired error guarantee.

**Lemma D.5.** *Incurring $\gamma^{\frac{1}{p-1}} \cdot f_1^{p-1} \cdot (n - f_1)$ error gives us $\gamma^{\frac{1}{p-1}} \cdot (n^p - F_p)$ total error.*

*Proof.* We have that $n^p - F_p \geq f_1^{p-1} \cdot (n - f_1)$ for $p \geq 2$. $n^p - F_p$ counts the number of $p$-tuples (allowing repetitions) in which not all entries of the tuple have the same value. The right hand side counts $p$-tuples in which all but one entry are equal to the value of highest frequency (i.e. $f_1$) and the last has a different value. We can see that the right hand side is a subset of the left hand side. Note that we can assume $p \leq \frac{\gamma}{2} \cdot n$ since $p$ is a constant. Since we said we could assume that $f_1 \geq \frac{\gamma^{\frac{1}{p-1}}}{2} \cdot n$, we have that $f_1 \geq p$. $\qquad\square$

Recall that we find our final approximation of $f_b'$ separately if our initial estimate shows that it is greater than $n/2$. Specifically, we instead subtract off $b'$ from $L_0^2$ and query it to get $w_2$. Then we set $f_b' = n - w_2$. We show that this does not incur too much error.

**Lemma D.6.** *If $f_1 \geq \frac{2n}{3}$, the error incurred from our estimate of $f_1$ is at most $\frac{\gamma^{\frac{1}{p-1}}}{3} \cdot f_1^{p-1} \cdot (n - f_1)$.*

*Proof.* Let us denote the distinct value that has frequency $f_1$ in $x$ as $b$. By Lemma D.4, we incur at most $3\varepsilon \cdot n$ error in estimating the frequency of $b$ using $\mathcal{S}_1$ (or at most $\varepsilon \cdot n$ error if $b = 0$). Since $f_1 \geq 2n/3$, the next largest frequency is at most $n/3$. Therefore we will not mistake another value for $b$. In addition we will estimate $f_b'$ to be at least $n/2$.

Since we correctly identify $b$ and correctly determine that its frequency is at least $n/2$, in our algorithm we will subtract off $b$ from $L_0^2$ and then query it to get $w_2$. Then we estimate the frequency as $n - w_2$.

Since $L_0^2$ is a linear sketch, "subtracting off $b$" means that we simulate updating all the entries of underlying vector $x$ by subtracting $b$ from them. Then querying $L_0^2$ will give us the number of entries of $x$ that are not equal to $b$.

By the properties of $L_0^2$, we incur at most $\Delta(f) = \varepsilon \cdot (n - f_1)$. Therefore, our total error is at most

$$(f_1 + \Delta(f))^p - f_1^p = \sum_{j=1}^{p} \left[ \binom{p}{j} f_1^{p-j} \Delta(f)^j \right] = \Delta(f) \sum_{j=1}^{p} \left[ \binom{p}{j} f_1^{p-j} \Delta(f)^{j-1} \right]$$

$$\leq \Delta(f) \sum_{j=1}^{p} \left[ \binom{p}{j} f_1^{p-1} \right] = \Delta(f) f_1^{p-1} \cdot 2^p$$

giving us the desired error. Note that our estimate of $f_1$ could have been $f_1 - \Delta(f)$ but we have $|(f_1 + \Delta(f))^p - f_1^p| \geq |(f_1 - \Delta(f))^p - f_1^p|$. $\qquad\square$

Let us now consider the case where we do not have $f_1 \geq 2n/3$ but in the algorithm we identify a value $v$ with estimated frequency $f_v' \geq n/2$. By Lemma D.4, we incur at most $3\varepsilon \cdot n$ error in estimating frequencies using $\mathcal{S}_1$ (or by $L_0^1$ for the value 0). So it must be that $f_v = \Theta(n)$, and by the same analysis as Lemma D.6 we get that this does not incur too much error.

We now show that estimating the values of frequency at least $\frac{\gamma^{\frac{1}{p-1}}}{2} \cdot (n - f_1)$ does not incur too much error. We denote a set $\mathcal{F}$ which contains every value of $x$ with frequency at least $\frac{\gamma^{\frac{1}{p-1}}}{2} \cdot (n - f_1)$.

**Lemma D.7.** *The error incurred from estimating $\sum_{i \in \mathcal{F}} f_i^p$ is at most $\frac{\gamma^{\frac{1}{p-1}}}{3} \cdot f_1^{p-1} \cdot (n - f_1)$ with probability at least $1 - \delta/8$.*

*Proof.* We first show how much error we incur by estimating the frequency of one value in $\mathcal{F}$. By Lemma D.2, we incur $\varepsilon(n - f_1)$ error in estimating the frequency if $w_2$ had no error in approximating $n - f_1$. By the correctness of $L_0^2$, $w_2$ is a $(1 \pm \varepsilon)$ multiplicative approximation to $n - f_1$. Therefore the total error in approximating $f_i$ with $f_{\mathcal{S}_2,i} \cdot w_2/t$ where $f_{\mathcal{S}_2,i}$ is the frequency of $i$ in $\mathcal{S}_2$ is at most $3\varepsilon \cdot (n - f_1)$.

By similar reasoning above, even if we choose $b$ incorrectly, we know by Lemma D.4 that $f_{b'} \geq f_b - 6\varepsilon \cdot n$. Therefore, subtracting off $b'$ to from $\mathcal{S}_2$ and estimate the frequencies of values in $\mathcal{F}$ by reweighing $\varepsilon$ still gets appropriate error. In

addition, note that because in the algorithm we add all values with estimate frequency at least $\frac{\gamma^{\frac{1}{p-1}}}{4} \cdot (n - f_1')$, we will put all values that are in $\mathcal{F}$ in $\mathcal{B}$ correctly. We now look at the error incurred in estimating all the frequencies of values in $\mathcal{B}$.

We denote $\Delta(f_i) = 3 \cdot \varepsilon \cdot (n - f_1)$. Let us consider all frequencies except $f_1$. We have that the error is at most

$$
\sum_{i \in \mathcal{B}, i > 1} [(f_i')^p - f_i^p] = \sum_{i \in \mathcal{B}, i > 1} [(f_i + \Delta(f_i))^p - f_i^p]
$$

$$
= \sum_{i \in \mathcal{B}, i > 1} \left[ \sum_{j=1}^{p} \left( \binom{p}{j} f_i^{p-j} \Delta(f_i)^j \right) \right] \leq \sum_{i \in \mathcal{B}, i > 1} \left[ \Delta(f_i) \sum_{j=1}^{p} \left( \binom{p}{j} f_i^{p-1} \right) \right]
$$

$$
\leq \sum_{i \in \mathcal{B}, i > 1} \left[ \Delta(f_i) \cdot 2^p \cdot f_i^{p-1} \right] = 2^p \cdot f_1^{p-1} \cdot \sum_{i \in \mathcal{B}, i > 1} \Delta(f_i).
$$

We will now show that $\sum_{i \in \mathcal{B}, i > 1} \Delta(f_i)$ is appropriately bounded. Note that $\sum_{i \in \mathcal{B}, i > 1} \Delta(f_i)$ is the sum of the errors in calculating the frequencies of values in $\mathcal{B}$ (except for $f_1$). Following the proof of Lemma D.2, we have $\mathbb{E}[f_i'] = r \cdot f_i$ and $\text{Var}[f_i'] \leq r \cdot f_i$. This gives us $\mathbb{E}\left[ \sum_{i \neq 1} f_i' \right] = r \cdot \sum_{i \neq 1} f_i$ and $\text{Var}[\sum_{i \neq 1} f_i'] \leq r \cdot \sum_{i \neq 1} f_i$ since the covariance of $f_j'$ and $f_k'$ for $j \neq k$ is negative. Recall that we have $\sum_{i \neq 1} f_i \leq n - f_1$ and that we set $r = \tilde{O}(\gamma^{-2})$. So, we can now apply Chebyshev's (with success amplification) to get that with probability at least $1 - \delta/8$ we have $\sum_{i \in \mathcal{B}, i > 1} \Delta(f_i) \leq \frac{\varepsilon}{2} \cdot (n - f_1)$.

If we had $f_1 \leq 2n/3$, we get error $\Theta(\varepsilon n)$ from estimating its frequency from $\mathcal{S}_1$ as proven by Lemma D.4. Since we know that $\Theta(n - \varepsilon n) \leq f_1 \leq 2n/3$, by re-weighing $\varepsilon$ we get appropriate error. □

We now deal with values $j$ such that $f_j \leq \frac{\gamma^{\frac{1}{p-1}}}{2} \cdot (n - f_1)$. We potentially do not approximate these frequencies. However, their contribution to $\sum f_i^p$ is low, and they give us small error as show below.

**Lemma D.8.** *The error incurred by not estimating values with frequency less than $\frac{\gamma^{\frac{1}{p-1}}}{2} \cdot (n - f_1)$ is at most $\frac{\gamma^{\frac{1}{p-1}}}{3} \cdot (n^p - F_p)$.*

*Proof.* We first observe that we have $\sum_{i \neq 1} f_i = n - f_1$. So, $\sum_{i \notin \mathcal{F}} f_i^p$ is greatest when there are $\frac{2}{\gamma^{\frac{1}{p-1}}}$ values each with frequency $\frac{\gamma^{\frac{1}{p-1}}}{2} \cdot (n - f_1)$. So this sum (and therefore the error we incur) is at most

$$
\sum_{i}^{2/\gamma^{\frac{1}{p-1}}} \left( \frac{\gamma^{\frac{1}{p-1}}}{2} \cdot (n - f_1) \right)^p \leq \frac{\gamma}{2^{p-1}} \cdot (n - f_1)^p.
$$

We have that $(n - f_1)^p \leq n^p - f_1^p$ so we are getting $\frac{\gamma}{2^{p-1}} \cdot (n^p - f_1^p)$ total error.

The quantity that we want to estimate is $n^p - f_1^p - \sum_{i > 1} f_i^p$. By Jensen's inequality we can see that

$$
n^p - f_1^p - \sum_{i > 1} f_i^p \geq n^p - f_1^p - \frac{(n - f_1)^p}{c}
$$

for some constant $c \geq 2$ since we have $\sum_{i > 1} f_1 = n - f_1$ and our summation is over at most $2/\gamma^{\frac{1}{p-1}}$ frequencies. Furthermore, we have that $n^p - f_1^p \geq (n - f_1)^p$. So, achieving $\frac{\gamma}{2^{p-1}} \cdot (n^p - f_1^p)$ gives us the desired error guarantee. □

Therefore, combining all the lemmas above gives the result.

## E. Proof of Theorem 1.5 (General Fingerprinting)

We now present our algorithm for general fingerprinting, general-fingerprinting-sketch (Algorithm 8) to prove Theorem 1.5. To utilize our general submodular maximization framework from Theorem 1.3, we need to provide a sketch that preserves queried values of the general fingerprinting function to within a $(1 \pm \gamma)$ factor. The general fingerprinting function receives as input a subset of the columns of $\mathbf{A}$ and outputs how many pairs of users they separate. We can therefore see that

maximizing this function gives us the desired output. Note that the general fingerprinting function is submodular since when adding a new column to a set $\mathcal{C}$ of columns, if this separates a pair of users that were previously not separated, then this column also separates that pair of users if added to some $T \subseteq \mathcal{C}$. It is also monotone since adding another column to $\mathcal{C}$ never decreases the function value.

Let us analyze the memory usage. We keep two $L_0$ sketches per column of $\boldsymbol{A}$. As per Theorem 1.3, we must set $\gamma = \varepsilon/k$ for our sketch. This makes the space of each $L_0$ sketch $\tilde{O}(k^2/\varepsilon^2)$. So the total space for all $d$ columns is $\tilde{O}(dk^2/\varepsilon^2)$. The space for each $\ell_0$ sampler is $\tilde{O}(\log^2 n)$, and we keep $\tilde{O}(dk^2/\varepsilon^2)$ of them giving us $\tilde{O}(dk^2/\varepsilon^2)$. Using Theorem 1.3, our total space is therefore $\tilde{O}(dk^3/\varepsilon^2)$.

For the update time, each update affects one column of $\boldsymbol{A}$, and therefore two $L_0$ sketches and $\tilde{O}(\gamma^{-2})$ $\ell_0$ samplers. So, the update time per sketch is $\tilde{O}(\gamma^{-2})$. As per Theorem 1.3, we will keep $k$ sketches so the total update time is $\tilde{O}(k/\gamma^2) = \tilde{O}(k^3/\varepsilon^2)$.

Now, we prove the correctness. As per our framework in Theorem 1.3, our result follows if we can show that our sketch provides $(1 \pm \gamma)$-approximations to all queried values to our general fingerprinting function with probability $O(1/(\text{poly}(d)k))$.

Upon a query to our function on a subset of columns $\mathcal{C}$, we return $g_S(\{\boldsymbol{S} \cdot \boldsymbol{a}_i\}_{i \in \mathcal{C}})$. To do this, for each type of sketch (both the $L_0$ sketch and the $\ell_0$-sampling sketch) for the columns of subset $\mathcal{C}$, we concatenate them and reduce them each to one column. Below, $(\boldsymbol{SA})_{\mathcal{C}}$ denotes the sketch of $\boldsymbol{A}$ restricted to the columns in $\mathcal{C}$.

**Lemma E.1.** *With probability $1 - 1/(\text{poly}(d)k)$, for any rows $x$ and $y$ in $(\boldsymbol{SA})_{\mathcal{C}}$ for sketch $\boldsymbol{S}$, they are distinct if and only if entry $x$ and $y$ of $[(\boldsymbol{SA})_{\mathcal{C}}]\boldsymbol{v}$ are distinct for random vector $\boldsymbol{v}$ with entries in $\{-\text{poly}(ndk), \text{poly}(ndk)\}$.*

*Proof.* Let us look at two rows of $\boldsymbol{B} = (\boldsymbol{SA})_{\mathcal{C}}$ that are distinct. We call these rows $\boldsymbol{B}_x$ and $\boldsymbol{B}_y$. Take $\boldsymbol{w}$ to be the vector that is formed from performing $\boldsymbol{B}_x - \boldsymbol{B}_y$. We first want to show that $\boldsymbol{w}^{\mathsf{T}}\boldsymbol{v} \neq 0$.

We have that $\boldsymbol{w}^{\mathsf{T}}\boldsymbol{v} = \boldsymbol{w}_1 \cdot \boldsymbol{v}_1 + \boldsymbol{w}_2 \cdot \boldsymbol{v}_2 + \cdots + \boldsymbol{w}_d \cdot \boldsymbol{v}_d$. Fixing the values of $\boldsymbol{v}_1$ through $\boldsymbol{v}_{d-1}$, there is only one value for $\boldsymbol{v}_d$ such that $\boldsymbol{w}^{\mathsf{T}}\boldsymbol{v} = 0$. Therefore, this "bad" event happens with probability at most $1/\text{poly}(ndk)$. Union bounding over all possible rows of $\boldsymbol{B}$, we have that with probability $1 - 1/\text{poly}(dk)$ if rows $x$ and $y$ of $\boldsymbol{B}$ for any $x, y$ are distinct then entries $x$ and $y$ of $\boldsymbol{Bv}$ are distinct.

To finish up the proof, we want to show that if rows $x$ and $y$ of $\boldsymbol{B}$ for any $x, y$ are identical, then entries $x$ and $y$ of $\boldsymbol{Bv}$ are identical. This is clearly true with probability 1. $\qquad\square$

Now, we are in the vector case. We claim that the rest of the work is done by passing in $\mathcal{S}_1$, $\mathcal{S}_2$, $\mathcal{S}_3$, and $\mathcal{S}_4$ into our sketch from Theorem 1.4 with $p = 2$. For each distinct item $i$ in the vector, we denote its frequency as $f_i$. As we can see, $\binom{n}{2} - \sum_i \binom{f_i}{2} = \frac{n^2 - F_2}{2}$ is the general fingerprinting function. This is because $\binom{n}{2}$ denotes all pairs of users and by subtracting off $\sum_i \binom{f_i}{2}$ we are subtracting off pairs of users that share identical values. Note the changes in the parameters of the input between here and in Theorem 1.4.

---

**Algorithm 5** Max-Coverage-LS ($n \times d$ matrix $\boldsymbol{A}$, $\varepsilon \in (0,1)$, $k$)

---

1: Set $\delta = (2 + \log d) \log \log_{1-\varepsilon} n$.
2: Set $\varepsilon = \varepsilon/8$.
3: Keep a $L_0$ sketch for each column of $\boldsymbol{A}$. Denote these as $L_0(z)$ for $z \in [d]$.
4: **for** $m = 1, 2, \ldots, \log n$ **do**
5:     {Run in parallel}
6:     Use a hash function to subsample each row from $\boldsymbol{A}$ with probability $1/2^m$. Call the subsampled matrix we consider in this $m^{\text{th}}$ run $\boldsymbol{A}'_m$.
7:     {We do not store $\boldsymbol{A}'_m$ explicitly. This means that in the $m^{\text{th}}$ parallel run we only consider updates to the subsampled rows that form $\boldsymbol{A}'_m$.}
8:     Set $b = O(\frac{k \log d}{\varepsilon^2})$.
9:     Set $t = O(\log \frac{d}{\varepsilon})$.
10:     **for** $i = 1, \ldots, t$ **do**
11:         Use a hash function to hash each row of $\boldsymbol{A}'_m$ to $b$ buckets in structure $\mathcal{C}_{m,i}$.
12:         {We do not store the rows of $\boldsymbol{A}'_m$ explicitly in structure $\mathcal{C}_{m,i}$. Rather, each bucket only considers updates to the rows that are hashed there.}
13:         **for** each bucket in $\mathcal{C}_{m,i}$ **do**
14:             If there are $r$ rows hashed to the bucket, denote the $r$ rows concatenated into a vector of length $rd$ as $\boldsymbol{v}$.
15:             **for** $q = 1, 2, \ldots, \log(rd)$ **do**
16:                 {Run in parallel}
17:                 Use a hash function to subsample each entry of $\boldsymbol{v}$ with probability $1/2^q$. Call the subsampled vector we consider in this $q^{\text{th}}$ run $\boldsymbol{v}_q$ {we again do not store $\boldsymbol{v}_q$ explicitly}.
18:                 Keep a $L_0$ sketch for $\boldsymbol{v}_q$. Denote it as $L_{0,q}$.
19:                 Keep a CountSketch structure with $O(\frac{d \log(1/\varepsilon)}{\varepsilon k})$ buckets for $\boldsymbol{v}_q$. Denote it as $CS_q$.
20:             **end for**
21:         **end for**
22:     **end for**
23: **end for**
24: Set the error probability for each $L_0$ sketch and CountSketch structure such that the total error across all of them is at most $1/\operatorname{poly}(d)$.
25: Upon an update, the $L_0$ sketches and CountSketch structures will handle it.
26: **Upon a query:**
27: **for** each $m \in [\log n]$ **do**
28:     Initialize $\boldsymbol{A}_{m,*}(k, \varepsilon)$.
29:     For each iteration $i \in [t]$, for each bucket in $\mathcal{C}_{m,i}$, take the smallest $q$ such that querying $L_{0,q}$ returns a number that is $O(\frac{d \log(1/\varepsilon)}{\varepsilon k})$ to be $q'$.
30:     Take $\mathcal{S}$ to be the set of rows of $\boldsymbol{A}'_m$ that have nonzero entries recovered by $CS_{q'}$ for any bucket in $\mathcal{C}_{m,i}$ for any iteration $i \in [t]$. Take $\mathcal{P}$ to be the set of these recovered nonzero entries.
31:     Let $h$ be a hash function that maps uniformly between $[0,1]$ the rows in $\mathcal{S}$.
32:     **while** the number of nonzero entries in $\boldsymbol{A}_{m,*}(k, \varepsilon)$ is less than $\frac{24d\delta' \log(1/\varepsilon) \log d}{(1-\varepsilon)\varepsilon^3}$ **do**
33:         Process the row $j$ that comes next in the ordering as determined by hash function $h$.
34:         **if** row $j$ has less than $\frac{d \log(1/\varepsilon)}{\varepsilon k}$ nonzero entries in $\mathcal{P}$ **then**
35:             Add all of the nonzero entries from row $j$ in $\mathcal{P}$ to $\boldsymbol{A}_{m,*}(k, \varepsilon)$.
36:         **else**
37:             Add $\frac{d \log(1/\varepsilon)}{\varepsilon k}$ of the nonzero entries from row $j$ in $\mathcal{P}$, chosen arbitrarily, to $\boldsymbol{A}_{m,*}(k, \varepsilon)$.
38:         **end if**
39:     **end while**
40: **end for**
41: Output $L_0(z)$ for $z \in [d]$ and $\boldsymbol{A}_{m,*}(k, \varepsilon)$ for $m \in [\log n]$.

---

**Algorithm 6** sketchy-submodular-maximization

---

1: Initialize $\mathcal{C} \leftarrow \emptyset$.
2: **while** $|\mathcal{C}| \leq k$ **do**
3: $\quad \mathcal{C} \leftarrow \mathcal{C} \cup \mathrm{argmax}_{c \in d \setminus \mathcal{C}} \, g_{\boldsymbol{S}}(\{\boldsymbol{S} \cdot a_i\}_{i \in c | \mathcal{C}})$.
4: **end while**
5: Return $\mathcal{C}$.

---

---

**Algorithm 7** p-Tuples-Sketch ($n \times 1$ vector $\boldsymbol{x}$, constant integer $p \geq 2$, $\gamma, \delta \in (0,1)$)

---

1: $\varepsilon \leftarrow \frac{\gamma^{\frac{1}{p-1}}}{16 \cdot 2^p}$.
2: Keep two independent $L_0$ sketches, $L_0^1, L_0^2$ of $\boldsymbol{x}$ each with $\delta' = \delta/8$ and $\varepsilon = \varepsilon$.
3: For $t = 2\varepsilon^{-2} \cdot \log(2(\delta/8)^{-1})$, keep $2t$ perfect $\ell_0$ samplers of $\boldsymbol{x}$. Denote the first $t$ as $\mathcal{S}_1$ and the others as $\mathcal{S}_2$.
4: Set $\delta'$ for each $\ell_0$ sampler s.t. the total probability of failure across them is at most $\delta/8$.
5: **Upon an update:**
6: The $L_0$ sketches and perfect $\ell_0$ samplers will handle updates.
7: **Upon a query:**
8: Initialize an empty set $\mathcal{B}$.
9: {Estimating the frequency of the highest frequency value, $b$.}
10: Query $L_0^1$ to get $w_1$. Set $b \leftarrow 0$ and $f_b' \leftarrow n - w_1$.
11: Estimate the frequency of a value using $\mathcal{S}_1$ by taking its frequency in $\mathcal{S}_1$ and scaling by $w_1/t$.
12: Find the value $v$ with highest frequency $f'$ in $\mathcal{S}$.
13: **If** $f' > f_b'$ **then** $b \leftarrow v$ and $f_b' \leftarrow f'$.
14: **If** $f_b' < \frac{3\gamma^{\frac{1}{p-1}}}{4} \cdot n$ **then** output $n^p$.
15: {If frequency of $b$ is large enough, estimate it separately for appropriate error.}
16: **if** $f_b' > \frac{n}{2}$ **then**
17: $\quad$ Subtract off value $b$ from $L_0^2$ and query it to get $w_2$.
18: $\quad$ Set $f_b' = n - w_2$.
19: **end if**
20: Add $(b, f_b')$ to $\mathcal{B}$.
21: Simulate update $x_i \leftarrow x_i - b$ for all $i \in [n]$ {$\mathcal{S}_2$ will update}.
22: Use $\mathcal{S}_2$ to get all values and their frequencies (take the frequency in $\mathcal{S}_2$ and scale by $w_2/t$).
23: **for** all values $v$ with frequency $f_v' \geq \frac{\gamma^{\frac{1}{p-1}}}{4}(n - f_b')$ **do**
24: $\quad$ Add $(v, f_v')$ to $\mathcal{B}$.
25: **end for**
26: Using all $z'$ tuples $(v, f_v') \in \mathcal{B}$, calculate $n^p - \sum_{j=1}^{z'}(f_j')^p$ and output.

---

---

**Algorithm 8** general-fingerprinting-sketch ($n \times d$ matrix $\boldsymbol{A}$, $\varepsilon \in (0,1), k \geq 0$)

---

1: $\gamma \leftarrow \varepsilon/k$.
2: **for** $j \in [d]$ **do**
3:     Maintain two $L_0$ sketches with $\varepsilon' = \gamma$ and $\tilde{O}(\gamma^{-2})$ perfect $\ell_0$ samplers for the $j^{\text{th}}$ column of $\boldsymbol{A}$.
4: **end for**
5: **To answer a query:**
6: The query will ask for the function value on a subset of columns $\mathcal{C}$.
7: For every $j \in \mathcal{C}$, take its first $L_0$ sketch and concatenate them into a matrix (each sketch is a column of the matrix). Denote the matrix as $\mathcal{S}_1$.
8: For every $j \in \mathcal{C}$, take its second $L_0$ sketch and concatenate them into a matrix. Denote the matrix as $\mathcal{S}_2$.
9: For each $j \in [d]$, we view the first half of its $\ell_0$ samplers as a uniform sampling vector $L_1$. We view the second half as vector $L_2$.
10: For every $j \in \mathcal{C}$, take its $L_1$ and concatenate them into a matrix $\mathcal{S}_3$.
11: For every $j \in \mathcal{C}$, take its $L_2$ and concatenate them into a matrix $\mathcal{S}_4$.
12: Reduce the column dimension of $\mathcal{S}_1, \mathcal{S}_2, \mathcal{S}_3$, and $\mathcal{S}_4$ by right multiplying by a random vector $v$ from $\{-\operatorname{poly}(ndk), \ldots, \operatorname{poly}(ndk)\}^{|\mathcal{C}|}$.
13: Run p-Tuples-Sketch (Algorithm 7) with $\mathcal{S}_1, \mathcal{S}_2, \mathcal{S}_3$, and $\mathcal{S}_4$, $\delta = 1/(\operatorname{poly}(d)k)$, $\gamma = \varepsilon/k$, and $p = 2$ to estimate $\frac{n^2 - F_2}{2}$.

---

