# OpenReview forum: "Maximum Coverage in Turnstile Streams with Applications to Fingerprinting Measures"
_ICML.cc/2025/Conference — ICML 2025 poster_

### Official Review · Reviewer_h96F · 2025-03-07

**Overall Recommendation:** 3

**Summary:**

This paper considers the maximum coverage problem, where there is a universe of $n$ elements and we are presented with $d$ subsets of these $n$ elements. The goal is to retain and output a set of $k$ subsets (where $k$ is a parameter given in advance) whose union covers as many items as possible. The exact version of this problem is NP hard, but it is known that a greedy algorithm, which runs in polynomial time, provides a $1-1/e$ approximation, and this is known to be tight assuming $P \neq NP$.

This paper considers the setting where the subsets are presented in a (single-pass) streaming fashion: each time we see one subset and the objective is to retain a set of $k$ subsets with approximation factor as close as possible to $1-1/e$ with as little space (memory) as possible. The main result is a single pass algorithm, in a turnstile setting where an element can be added or removed to a single subset in each step. The space complexity for $1-1/e-\epsilon$ approximation is roughly $d / \epsilon^3$, and the update time is only polylogarithmic in $n$. The authors also provide applications to fingerprinting, where the goal is to find sets of users satisfying certain "minimally intersecting" constraints.

**Claims And Evidence:**

This is a theoretical paper and the claims are proved.

Can you provide a reference that $1-1/e$ approx factor is tight? (This was mentioned without proof in the introduction)

**Essential References Not Discussed:**

None that I know of.

**Experimental Designs Or Analyses:**

This is a theoretical paper, and the focus is on proving theorems, not on experimentation. That said, the authors do carry out a number of relevant experiments, which I think is more than enough for such a theoretically-oriented paper. I found the soundness of the experiments adequate, especially given the relatively limited baseline for this problem.

**Methods And Evaluation Criteria:**

Yes - solid experimental benchmarking (see relevant bullet below).

**Other Comments Or Suggestions:**

See weaknesses discussion above.

**Other Strengths And Weaknesses:**

This is a solid paper providing strong results for the maximum coverage problem, with interesting applications to fingerprinting. It is probably worthy of acceptance.

There are no major weaknesses. Perhaps one minor weakness is that this paper is written in a relatively dry manner, and seems mostly intended for experts. A more detailed subsection on proof techniques (as part of the intro) is also missing and should be added.

**Questions For Authors:**

None

**Relation To Broader Scientific Literature:**

There is not a lot of research on this problem in the streaming domain, but the authors do identify the most relevant work and benchmark against it.

**Theoretical Claims:**

The theoretical claims are strong and seem reasonable. I did not verify their correctness.

---

> ### Author Rebuttal · Authors · 2025-04-01
>
> We thank the reviewer for their review and encouraging comments. We address them below.
> Here is a reference showing that a 1-1/e approximation (unless P = NP) in polynomial time is tight [1]. We will be sure to include this citation in the next version of the paper.
> We will also be sure to include a more detailed subsection on our proof techniques in the introduction. In particular, we will give an algorithm and proof sketch and more details on why we can accommodate deletions.
>
> [1] Uriel Feige. 1998. A threshold of ln n for approximating set cover. J. ACM 45, 4 (July 1998), 634–652. https://doi.org/10.1145/285055.285059
>
> Please let us know if there is anything else that would be helpful to address or clarify.

---

### Official Review · Reviewer_gY8Y · 2025-03-12

**Overall Recommendation:** 3

**Summary:**

The paper considers the maximum coverage problem in the turnstile model.
The offline problem considers $d$ subsets from a universe $[n]$, and the goal is to output $k < d$ subsets such that the union of the sets contains the largest possible number of items from $[n]$.
This can also be expressed in matrix notation where $A\in\mathbb{R}^{n\times d}$: each column is a subset, each row is an item from the universe.
$A_{ij}\neq 0$ means that item $i$ is present in subset $j$.
In the turnstile model, the matrix is gradually constructed by updates of the form $(i, j, \pm c)$, modifying $A_{ij}$ by adding or subtracting $c$.
The challenge is to continuously solve the maximum coverage problem on $A$ as it gets updated, in low space.
Without the restriction on space, the problem is simple, as $A$ can be stored uncompressed and running a greedy algorithm achieves a tight $1-1/e$ relative approximation (assuming $P\neq NP$), and so we could simply run a greedy algorithm at each step.
With a restriction on space, the problem is harder.

Previous work achieved $(1-1/e -\epsilon)$-relative approximation for only set-arrival in space $\tilde{O}(d/\epsilon^2)$ (McGregor & Vu, 2018).
Bateni et al. (2017), achieved the same error guarantee for item-arrival in space $\tilde{O}(d/\epsilon^3)$.
Neither work supported deletions.
This paper gives a streaming algorithm that also supports item-deletion, for the same error guarantee, in $\tilde{O}(d/\epsilon^3)$, matching the guarantees of Bateni et al. (2017) for this harder setting (Theorem 1.1).

The algorithm proposed in this paper uses Bateni et al. (2017) as the starting point.
There the idea is that a matrix $A_*$ on a smaller universe can be constructed by carefully subsampling $A$, and that greedily running $k$-cover on $A_*$ gives the right error guarantee.
In this paper, they give an algorithm (Algorithm 1) for constructing $A_*$ assuming a fixed $A$ is given up front.
They proceed to show how (multiple) $A_*$ can be built from $A$ using a linear sketch (Algorithm 5), internally based on CountSketch and a $L_0$ sketch.
As CountSketch and the $L_0$ sketch are linear sketches, and linear sketches support both additions and deletions, so does Algorithm 5.
When outputting an answer, they run $k$-cover on each of the $\log n$ different (corresponding to different subsampling rates of the rows of $A$) $A_*$, and choose the best output based on the $L_0$ sketch.

The paper also considers additional problems in turnstile streams.
The authors reduce targeted fingerprinting to maximum coverage, thereby improving over an existing algorithm in Gulyas et al. (2016) (Corollary 1.2), reducing space (from $O(nd)$ to $\tilde{O}(d/\epsilon^3)$) and time (from $O(knd)$ over all updates, to $\tilde{O}(1)$ per update).
For general fingerprinting, the authors design a new linear sketch for computing the complement of the frequency moment of a dataset, $n^p - F_p$ (Theorem 1.4), which they use for achieving a $(1-1/e-\epsilon)$-relative approximation for general fingerprinting in space $O(dk^3/\epsilon^2)$ and time.

The paper includes experiments on the fingerprinting results run on public datasets, comparing against Gulyas et al. (2016).
It is demonstrated that the new algorithms are more time and space efficient (Figure 1, 2 and 4), but at a cost in utility (Figure 2,3 and 5).
The authors also demonstrate that their general fingerprinting algorithm can be used for dimensionality reduction, with the use case of $k$-means.

## update after rebuttal

The authors provided good answers to my questions in their rebuttal. I stand by my score.

**Claims And Evidence:**

Yes.

**Essential References Not Discussed:**

I am not familiar with any missing work.

**Experimental Designs Or Analyses:**

The experiments performed in the paper make sense to me.

**Methods And Evaluation Criteria:**

Yes.

**Other Comments Or Suggestions:**

N/A.

**Other Strengths And Weaknesses:**

Strengths:
1. It is overall a well-written paper.
2. The contribution of expanding streaming maximum coverage to also work for deletions strikes me as meaningful, and I personally find the techniques interesting.
3. The empirical results make a good case for the improved time and space complexity (at a cost in utility).

Weaknesses:
1. It is hard to infer what was known before on fingerprinting from reading the paper. For targeted fingerprinting Corollary 1.2 and its preceding discussion seem to indicate that this paper improves the space and time complexity, but it is not clear to me if it achieved the same error guarantee. For general fingerprinting it also hard to infer how Theorem 1.5 compares to prior work (Gulyas et al., 2016).
2. Related to the above, it seems that the baseline always achieves better error in the experiments, but that this should be the case is not clear from the text.
3. Minor quibble: the figures could be better formatted/scaled.

**Questions For Authors:**

I ask the following questions to better understand the contributions made in this paper.
I do not expect to drastically change my score based on these responses in isolation.

Questions:
1. You seem to incur a factor of $O(\log n)$ in the space from storing $O(\log n)$ different versions of $A_*$ for the result in Theorem 1.1. Do Bateni et al. (2017) and McGregor & Vu (2018) also incur a logarithmic dependence on $n$?
2. What error guarantee do Gulyas et al. (2016) achieve for targeted and general fingerprinting? Do they get a $(1-1/e)$-approximation for both?
3. Is the space/time complexity of Gulyas et al. (2016) the same across general and targeted fingerprinting?

**Relation To Broader Scientific Literature:**

To the best of my knowledge, the results are properly contextualized.

**Theoretical Claims:**

I skimmed the appendix, but did not check any proof carefully.

---

> ### Author Rebuttal · Authors · 2025-04-01
>
> We thank the reviewer for their thorough review and comments. We address the comments and questions below.
>
> We will be sure to scale and format the figures more appropriately in the next version of the paper.
>
> Concerning the $O(\log n)$ factors in Theorem 1.1: Yes both Bateni et al. (2017) and McGregor & Vu (2018) incur $O(poly \log n)$ factors.
>
> Concerning the questions on prior work (Gulyas et al., 2016):
> (Gulyas et al., 2016) shows that max coverage can be reduced to fingerprinting proving that fingerprinting is NP-hard. In addition, it is known that the best approximation factor for maximum coverage in polynomial time is $1-1/e$. This is the approximation that (Gulyas et al., 2016) achieves, although it is not proven formally. For both targeted and general fingerprinting their algorithm takes O(nd) space and O(knd) time. In our paper we achieve a $1-1/e-\epsilon$ approximation for input $\epsilon \in (0,1)$, making our approximation near-optimal. In our paper we focus on optimizing the space and time bounds for maximum coverage, giving as a corollary a $\tilde{O}(d/\epsilon^3)$ space and $\tilde{O}(1)$ update time algorithm for targeted fingerprinting. This removes the dependence on $n$ and gives a total time nearly linear in the length of the stream. For general fingerprinting, we achieve $\tilde{O}(dk^3/\epsilon^2)$ space. We note that in these big data settings we usually assume that $n$ is much larger than $d$ and therefore aim to remove the space dependence on $n$. We will be sure to make this more clear in the next version of the paper.
>
> For the experiments, the baseline always does achieve a better error. We will make sure this is clearer in the text. We note that it is expected that the baseline achieves better error - the baseline is a slightly optimized version of the classical greedy maximum coverage algorithm which is known to achieve optimal error for algorithms using polynomial time. However, as noted in our experiments our algorithms do not suffer much in accuracy in comparison and greatly increase the time efficiency.
>
> Please let us know if there is anything else that would be helpful to address or clarify.

---

### Official Review · Reviewer_TbWm · 2025-03-12

**Overall Recommendation:** 4

**Summary:**

The paper introduces a linear sketch for the maximum coverage problem that supports both insertions and deletions of item-feature pairs under the turnstile streaming model. This sketch improves on previous work that considers insertion-only streams, or which support only the insertion or deletion of entire subsets rather than individual items from subsets. The main application of this sketch considered in the paper is the fingerprinting problem from the privacy literature, where the task is to select a subset of features that best distinguishes either a single user (i.e., the "targeted" case), or all pairs of users (the "general" case). As part of the machinery to apply the sketch to general fingerprinting, the paper describes an additional sketch for the complements of frequency moments. The empirical evaluation compares the runtime and accuracy of the proposed sketching approach to the fingerprinting method described in Gulyas et al. (2016). Additionally, an application to accelerate k-means via dimensionality reduction is also explored.

**Claims And Evidence:**

The claimed construction of a maximum coverage sketch supporting insertions and deletions is supported by the theoretical results. The claim that the sketching approach improves on the fingerprinting method from Gulyas et al. (2016) is supported by the run time and accuracy measurements in the evaluation.

**Essential References Not Discussed:**

-

**Experimental Designs Or Analyses:**

-

**Methods And Evaluation Criteria:**

The fingerprinting evaluation compares the proposed sketch against the method from Gulyas et al. (2016) on the UCI Adult and Census datasets. Two limitations of this evaluation are that: (1) it does not compare the proposed sketch against other sublinear space algorithms for maximum coverage like that of Bateni et al. (2017), which should also be applicable to the fingerprinting problem, and that (2) the evaluation does not touch on the proposed sketch's support for item deletions, which is the main differentiator of this approach from previous work. With respect to (1), one question that should ideally be addressed is: what is the penalty that is incurred vs. prior work in terms of the sketch's memory/accuracy tradeoff due to the added support for deletions?

**Other Comments Or Suggestions:**

Typos:
- L213-214 should read "This process continues until $A_*$ contains ... So, $A_*$ is ...".
- Algorithm 1, L2 should read "$\varepsilon = \epsilon / 8$"

**Other Strengths And Weaknesses:**

Strengths:
- The theoretical contributions of the paper are strong. The sketch constructions presented in the paper and the accompanying proofs of correctness are nontrivial and will be of interest to the broader streaming algorithms community.

Weaknesses:
- The empirical evaluation of the proposed sketch is limited, as detailed above.
- The figures in the evaluation section should be made more legible. The font size in the plots is small, and there is plenty of whitespace that can be filled with larger figures.

**Questions For Authors:**

1. Can you clarify how the sketch from Bateni et al. (2017) fails to support deletions?
2. The max coverage sketch is constructed using several Count Sketch and L0 sketches. What practical recommendations do you have for selecting the parameters for these component sketches?
3. The evaluation section of the paper focuses on small values of $k$. Can you comment on how the algorithm scales to larger values of k, e.g. for dimensionality reduction from 1000s to 100s of features?

**Relation To Broader Scientific Literature:**

The paper's contributions are related to the literature on sketching algorithms for performing analyses of data streams using sublinear space. For the maximum coverage problem, the proposed sketch improves on the results of Bateni et al. (2017) by supporting both insertions and deletions while maintaining the same memory cost up to polylog factors.

**Theoretical Claims:**

I checked the proofs of the main theorems at a high level and did not identify any correctness issues. I did not verify the proofs in detail.

---

> ### Author Rebuttal · Authors · 2025-04-01
>
> We thank the reviewer for their thorough and encouraging review. We address their comments and questions below.
>
> Regarding the experimental evaluation:
> We chose Gulyas et al. (2016) as the baseline primarily because it represents the standard offline approach for the fingerprinting application, allowing us to demonstrate the speedup achieved by our sketch. We did not include item deletions in our evaluation because the baseline (prior work Gulyas et al. (2016)) is not a streaming algorithm. Therefore, it does not support updates of any kind including deletions and requires direct access to the entire input (which in this case is input $n \times d$ matrix $A$). Regarding the penalty of adding support for deletions as compared to prior streaming work - our algorithm achieves the same asymptotic space complexity of $\tilde{O}(d/\epsilon^3)$ and near-optimal approximation factor $(1-1/e-\epsilon)$ as the insertion-only algorithm of Bateni et al. (2017). While constants might differ in practice, theoretically we match the bounds while offering broader functionality.
>
> We will be sure to improve the formatting and legibility of the plots in the next version of the paper. We thank the reviewer for also pointing out some typos - we will be sure to correct them.
>
> Regarding the question on the sketch from Bateni et al. (2017):
> The sketch from Bateni et al. (2017) fails to accommodate deletions. Up to $\tilde{d/\epsilon^3}$ edges, the sketch requires $d/\epsilon$ nonzero elements per item/row of input matrix $A$. In particular, in their Algorithm 2 where they implement this sketch in the streaming setting, they keep the first $d/\epsilon$ nonzeros for some row $r$ and discard all the nonzeros for that row $r$ that come in the stream after this. However, in a stream that has deletions, some subset (or all) of these first $d/\epsilon$ nonzeros could be deleted, leaving the sketch with no nonzeros for this row. It is not clear how to get around this issue with the algorithm given in the paper of Bateni et al. (2017).
>
> Regarding practical recommendations for selecting parameters for CountSketch and L0 sketches: While our theoretical analysis establishes the necessary parameters to achieve the desired accuracy and failure probability, practical implementations often require less space and time to obtain good approximations. Our experiments support this observation. We set the number of buckets and other sketch-related constants in our implementation to fixed values, which we either varied for experimental analysis or fine-tuned. These values were lower than the theoretical requirements.
>
> Regarding the question about how the algorithm scales:
> In our experiments we focus on small values of $k$ due to compute limitations. We note that theoretically the only algorithm whose space complexity depends linearly on $k$ is the one for general fingerprinting. In particular, the space of the core max coverage sketch is independent of $k$. In addition, our algorithms are designed for big data settings and we expect our algorithms to scale well with huge values of $n$, $d$, and $k$. Furthermore, as $n,d,k$ grow, we expect the accuracy guarantees to hold while the runtime advantage over the $O(knd)$ baseline becomes more pronounced. Evidence of this can be seen in our speed-up for the larger dataset versus the smaller dataset.
>
> Please let us know if there is anything else that would be helpful to address or clarify.

---

### Official Review · Reviewer_ZvF7 · 2025-03-17

**Overall Recommendation:** 4

**Summary:**

This paper studies the maximum coverage problem in the data stream model and give the first turnstile algorithm, i.e., allowing both insertion and deletion, for this problem with space complexity almost match previous insertion-only streaming algorithms.

**Claims And Evidence:**

Yes, the claims made in the submission are supported by clear and convincing evidence.

**Essential References Not Discussed:**

No

**Experimental Designs Or Analyses:**

The experimental studies are not the strength of this work. But it made a nice complement to the theoretical results.

**Methods And Evaluation Criteria:**

Yes

**Other Comments Or Suggestions:**

NA

**Other Strengths And Weaknesses:**

Strength:
1. The main theoretical results are interesting to the streaming literature. In addition to minimizing the space complexity, update time is optimized as well.
2. Similar techniques are also applicable to fingerprinting problem.
3. Complementary experimental studies show that the proposed algorithms are competitive.

Weakness:
1. I feel that the techniques used are not too deep.
2. The problems studied do not have much "learning flavor" (although I have seen similar papers published in top machine learning venues). I think the author could adjust the writing to make it more machine learning oriented.

**Questions For Authors:**

NA

**Relation To Broader Scientific Literature:**

The key contributions of the paper is presenting the first turnstile streaming algorithms for the max coverage problem and some related problem. Linear sketching techniques are well-studied, but their applications to the problems studied in this paper are missing in the literature.

**Theoretical Claims:**

I have checked the correctness of key proofs. The proofs seem correct to me.

---

> ### Author Rebuttal · Authors · 2025-04-01
>
> We thank the reviewer for their review and encouraging comments. We will be sure to adjust the writing, in particular the introduction, to make it more machine learning oriented in the next version of the paper. For example, we will expand on the applications to sensor placement, influence maximization, and engagement maximization which are popular problems in machine learning and have been referenced in other machine learning papers [1, 2].
>
> [1] Zhou, H., Huang, L., and Wang, B. “Improved Approximation Algorithms for k-Submodular Maximization via Multilinear Extension.” ICLR 2025.
>
> [2] Tajdini, A., Jain, L., and Jamieson, K. “Nearly Minimax Optimal Submodular Maximization
> with Bandit Feedback.” NeurIPS 2024.
>
> Please let us know if there is anything else that would be helpful to address or clarify.

---

### Decision · Program_Chairs · 2025-05-01

**Decision:**

Accept (poster)

**Comment:**

This paper studies the maximum coverage problem and presents a streaming algorithm to solve this problem in the turnstile model, where updates to insert or delete an item arrive one by one. The proposed algorithm uses polylog n update time.

From the initial reviews, all reviewers were in favor of accepting the paper. Several points to improve the clarity were brought up in reviews:
* limitations of the evaluation: comparisons with other sublinear space algorithms for max coverage and evauating the sketch's support for item deletions. In general, the reviewer hoped to see the question "what is the penalty that is incurred vs. prior work in terms of the sketch's memory/accuracy tradeoff due to the added support for deletions?" addressed.

* More discussion of the literature on fingerprinting that appropriately contextualizes where this contribution lies in the literature. One reviewer wrote: "For targeted fingerprinting Corollary 1.2 and its preceding discussion seem to indicate that this paper improves the space and time complexity, but it is not clear to me if it achieved the same error guarantee. For general fingerprinting it also hard to infer how Theorem 1.5 compares to prior work (Gulyas et al., 2016)."

* Several reviewers commented that the formatting and legibility of the figures should be improved for the final camera ready version.

We encourage the reviewers to revise the final version of the manuscript based on the reviewer comments.